# Early Release Science of the exoplanet WASP-39b with JWST NIRSpec PRISM

Z. Rustamkulov[1✉], D. K. Sing[1,2], S. Mukherjee[3], E. M. May[4], J. Kirk[5,6], E. Schlawin[7], M. R. Line[8], C. Piaulet[9], A. L. Carter[3], N. E. Batalha[10], J. M. Goyal[11], M. López-Morales[5], J. D. Lothringer[12], R. J. MacDonald[13,14], S. E. Moran[15], K. B. Stevenson[4], H. R. Wakeford[16], N. Espinoza[17], J. L. Bean[18], N. M. Batalha[3], B. Benneke[9], Z. K. Berta-Thompson[19], I. J. M. Crossfield[20], P. Gao[21], L. Kreidberg[22], D. K. Powell[23], P. E. Cubillos[24], N. P. Gibson[25], J. Leconte[26], K. Molaverdikhani[27,28], N. K. Nikolov[17], V. Parmentier[29,30], P. Roy[9], J. Taylor[31], J. D. Turner[14], P. J. Wheatley[32,33], K. Aggarwal[34], E. Ahrer[32,33], M. K. Alam[21], L. Alderson[16], N. H. Allen[2], A. Banerjee[35], S. Barat[36], D. Barrado[37], J. K. Barstow[35], T. J. Bell[38], J. Blecic[39,40], J. Brande[20], S. Casewell[41], Q. Changeat[17,42,43], K. L. Chubb[44], N. Crouzet[45], T. Daylan[46], L. Decin[47], J. Désert[36], T. Mikal-Evans[22], A. D. Feinstein[18,35], L. Flagg[14], J. J. Fortney[3], J. Harrington[48], K. Heng[27], Y. Hong[14], R. Hu[49,50], N. Iro[51], T. Kataria[49], E. M.-R. Kempton[52], J. Krick[53], M. Lendl[54], J. Lillo-Box[37], A. Louca[45], J. Lustig-Yaeger[4], L. Mancini[22,24,55], M. Mansfield[7], N. J. Mayne[56], Y. Miguel[45,57], G. Morello[58,59,60], K. Ohno[3], E. Palle[58], D. J. M. Petit dit de la Roche[54], B. V. Rackham[61,62], M. Radica[9], L. Ramos-Rosado[2], S. Redfield[63], L. K. Rogers[64], E. L. Shkolnik[8], J. Southworth[65], J. Teske[21], P. Tremblin[66], G. S. Tucker[67], O. Venot[68], W. C. Waalkes[69], L. Welbanks[8], X. Zhang[70] & S. Zieba[22,45]

Transmission spectroscopy[1–3] of exoplanets has revealed signatures of water vapour, aerosols and alkali metals in a few dozen exoplanet atmospheres[4,5]. However, these previous inferences with the Hubble and Spitzer Space Telescopes were hindered by the observations' relatively narrow wavelength range and spectral resolving power, which precluded the unambiguous identification of other chemical species—in particular the primary carbon-bearing molecules[6,7]. Here we report a broad-wavelength 0.5–5.5 μm atmospheric transmission spectrum of WASP-39b[8], a 1,200 K, roughly Saturn-mass, Jupiter-radius exoplanet, measured with the JWST NIRSpec's PRISM mode[9] as part of the JWST Transiting Exoplanet Community Early Release Science Team Program[10–12]. We robustly detect several chemical species at high significance, including Na ($19\sigma$), $H_2O$ ($33\sigma$), $CO_2$ ($28\sigma$) and CO ($7\sigma$). The non-detection of $CH_4$, combined with a strong $CO_2$ feature, favours atmospheric models with a super-solar atmospheric metallicity. An unanticipated absorption feature at 4 μm is best explained by $SO_2$ ($2.7\sigma$), which could be a tracer of atmospheric photochemistry. These observations demonstrate JWST's sensitivity to a rich diversity of exoplanet compositions and chemical processes.

We observed one transit of WASP-39b on 10 July 2022 with JWST's Near InfraRed Spectrograph (NIRSpec)[9,13], using the PRISM mode, as part of the JWST Transiting Exoplanet Community Early Release Science Program (ERS Program 1366) (PIs: Natalie Batalha, Jacob Bean, Kevin Stevenson)[10,11]. These observations cover the 0.5–5.5 μm wavelength range at a native resolving power of 20–300. WASP-39b was selected for this JWST-ERS Program because of previous space- and ground-based observations revealing strong alkali metal absorption and several prominent $H_2O$ bands[4,6,14–16], suggesting a strong signal-to-noise ratio could be obtained with JWST. However, the limited wavelength range of existing transmission spectra (0.3–1.65 μm, combined with two wide photometric Spitzer channels at 3.6 and 4.5 μm) left several important questions unresolved. Previous estimates of WASP-39b's atmospheric metallicity—a measure of the relative abundance of all gases heavier than hydrogen or helium—vary by four orders of magnitude[6,16–20]. Accurate determinations of metallicity can explain formation pathways and

provide greater insight into the planet's history[21]. The JWST NIRSpec PRISM observations we present here offer a more detailed view into WASP-39b's atmospheric composition than has previously been possible (see ref. [21] for an initial infrared analysis of these data).

We obtained time-series spectroscopy over 8.23 h centred around the transit event to extract the wavelength-dependent absorption by the planet's atmosphere—that is, the transmission spectrum, which probes the planet's day-night terminator region near millibar pressures. We used NIRSpec PRISM in Bright Object Time Series (BOTS) mode. WASP-39 is a bright, nearby, relatively inactive[22] G7 type star with an effective temperature of 5,400 K (ref. [8]). WASP-39's J-band magnitude of 10.66 puts it near PRISM's saturation limit, which allows us to test the effects of saturation on the quality of the resulting science compared to past measurements (Methods).

In our baseline reduction using Fast InfraRed Exoplanet Fitting for Lightcurves (FIREFLy)[23], we perform calibrations on the raw data using

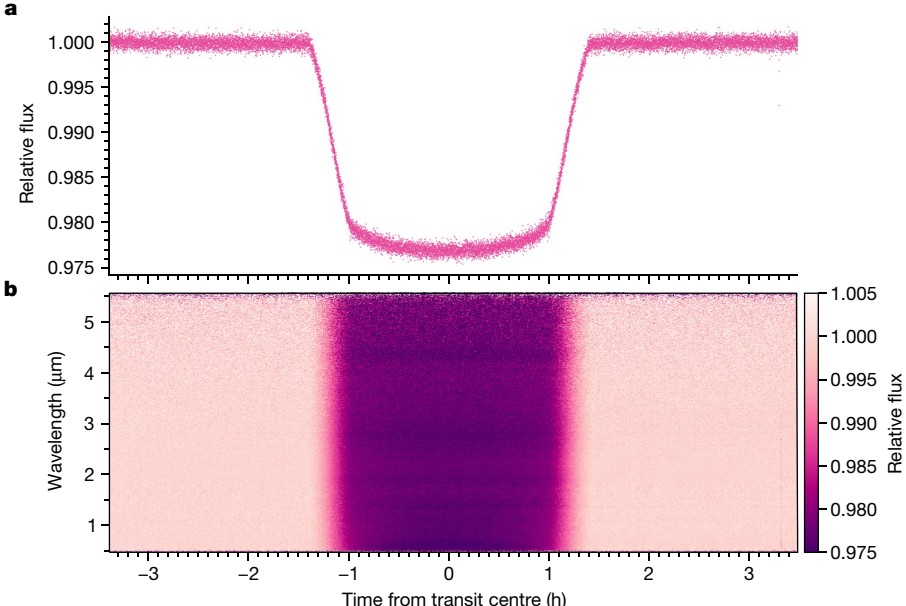

**Fig. 1 | The light curve of WASP-39b observed by JWST NIRSpec PRISM.**
**a**, The normalized white light curve created by integrating over all wavelengths using the FIREFLy reduction. **b**, The binned time series (with 30 integrations per time bin) of the relative flux for each wavelength. A constant 200 ppm per hour linear trend through time has been removed from the white light curve and each spectral channel for visual clarity.

the jwst Python pipeline[12] and then identify and correct for bad pixels and cosmic rays. We mitigate the $1/f$ noise[9] at the group level rather than the integration level to ensure accurate slope fitting, which we find to be a crucial step for NIRSpec PRISM observations with few groups per integration.

We bin the resulting spectrophotometry in wavelength to create 207 variable-width spectral channels with roughly equal counts in each. Figure 1 shows the FIREFLy white and spectrophotometric light curves at this step in the top panel. Several absorption features are visible by-eye as darker horizontal stripes within the transit region in the two-dimensional (2D) light curve (Fig. 1), demonstrating the high quality of the raw spectrophotometry achieved by the PRISM observing mode.

To extract the atmosphere's transmission spectrum, we fit the planet's transit depth in each wavelength bin using a limb darkened transit light curve model using the Python-based Levenburg–Marquardt least-squares algorithm lmfit[24]. The light curves show a typical photometric scatter of 0.2–1.2% per integration (1.36 s each), and the typical transit depth uncertainties vary between 50 and 200 parts per million (ppm), which is in line with near-photon-limited precision (Methods). Although we successfully measure fluxes in the saturated regions (0.8–2.3 μm), because of the lower number of groups used per integration here (1–3) the measured count rates may be adversely affected. We do not find excess red noise in the saturated channels themselves, however, we notice large point-to-point scatter in the transit depths, which required wider wavelength binning to better match previous Hubble Space Telescope (HST) observations. Figure 2 highlights representative transit light curves spanning the entire wavelength range. These data are binned into wider wavelength channels than those used for the final transmission spectrum for ease of presentation. Light curve systematics have not been removed from these data, demonstrating the unprecedented stability and precision of the PRISM observing mode.

We also compared the results from the FIREFLy reduction to three other independent reductions that use different treatments for the saturated region of the detector, limb darkening and various detector systematics (Methods). All four reductions obtain consistent results. Figure 3 shows a comparison of the four reductions. The consistency provides confidence in the accuracy of derived atmospheric parameters, demonstrating that any residual systematics are minimal and do

not strongly bias results for NIRSpec PRISM observations. The transmission spectrum also agrees well with previous measurements from ground-based telescopes[15,16] as well as HST and Spitzer[6] within error (Fig. 3), indicating that we can reliably recover a spectrum at these levels of saturation. These PRISM observations offer high-quality data from 0.5–5.5 μm, with minimal contributions from systematics and at precisions generally near the photon limit (Methods). Although recovery of the saturated region (0.9–1.5 μm) is possible, caution is warranted when interpreting this portion of the spectrum (Methods). Future PRISM observations of similarly bright targets should therefore carefully consider whether saturating the spectrum is an appropriate choice for a given planet, or whether building the wavelength coverage from several transits with different complementary modes is preferable.

The transmission spectrum of WASP-39b from the FIREFLy reduction is shown in Fig. 4. We select the FIREFLy reduction to be our baseline reduction, but comparable results are achieved with the three other reductions presented in this work (Methods). We interpret the spectrum with grids of one-dimensional (1D) radiative–convective–thermochemical equilibrium (RCTE) models (postprocessed with some more gases (Methods)), with a representative best-fitting model transmission spectrum shown in Fig. 4, along with opacity contributions from atoms, molecules and grey clouds. We detect the presence of $H_2O$ by means of four pronounced independent bands ($33\sigma$, 1–2.2 μm), a prominent $CO_2$ feature at 4.3 μm ($28\sigma$), Na at 0.58 μm ($19\sigma$), a CO absorption band at 4.7 μm ($7\sigma$) and a grey cloud ($21\sigma$). We do not observe any significant $CH_4$ absorption (expected at 3.3 μm), despite predictions of its presence for atmospheres at approximately solar metallicity and place a $3\sigma$ upper limit of $5 \times 10^{-6}$ on the $CH_4$ volume mixing ratio between 0.1 and 2 mbar. We also observe a relatively narrow absorption feature at 4.05 μm (roughy $2.7\sigma$), which we attribute to $SO_2$—a potential tracer for photochemistry[25–27]—after an extensive search across many possible opacity sources (Methods). Using a Bayesian approach described in the Methods section, we calculate that the volume mixing ratio of $SO_2$ needed to explain this feature is $10^{-5}$. The potential $SO_2$ feature is also observed at higher resolutions with JWST NIRSpec G395H (ref. [28]), adding confidence that the feature first reported as an unknown absorber[29] is a genuine feature of the planet's atmosphere. With Na detected in the atmosphere, the alkali metal, K, is also expected at

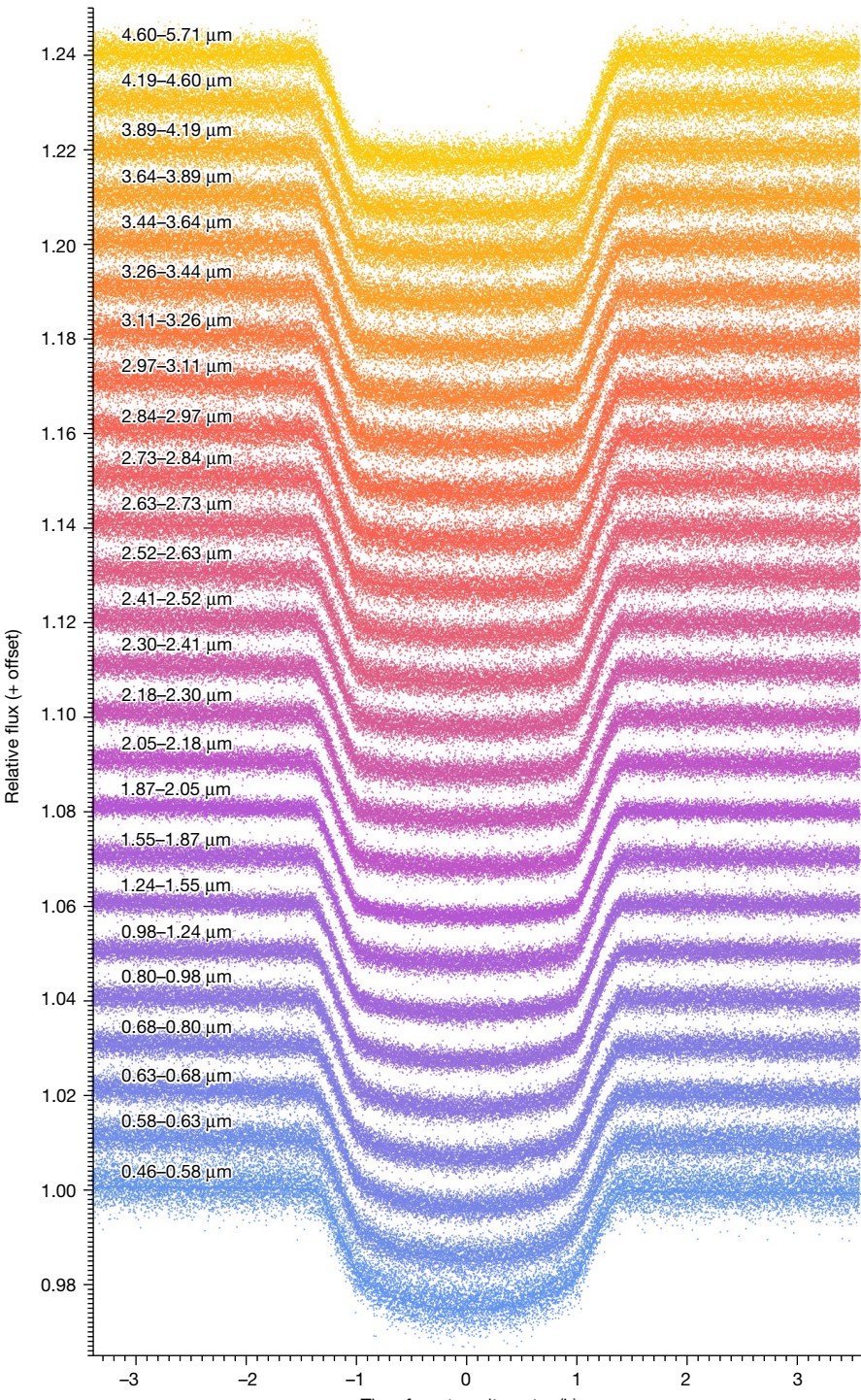

**Fig. 2 | Normalized spectrophotometric light curves for the JWST-PRISM transit of WASP-39b.** The light curves were created by summing over wide wavelength channels (wavelength ranges indicated on the plot). Overplotted on each light curve are their best-fit models, which include a transit model and detector systematics. Light curve systematics have not been removed from the data.

optical wavelengths[14] although not detected. However, the resolution covering the narrow K absorption doublet in the optical is low, which may be preventing detection. This might also be because of detector saturation in the wavelength range where K absorption is expected. We also do not detect the presence of $H_2S$ in the atmosphere. We note that although the best-fitting models shown in Figs. 3 and 4 have some $CH_4$, $H_2S$ and K signatures, these species are not favoured by the data to the level of a detection. We determine the single best-fitting atmospheric metallicity, C/O ratio and grey cloud opacity to be ten times solar, 0.7

and $\kappa_{cld} = 10^{-2.07}$ cm$^2$ g$^{-1}$, respectively. A detailed discussion on these best-fitting parameters is presented in the Methods section.

JWST/NIRSpec PRISM's power to constrain many chemical species in hot giant planet atmospheres provides new windows into their compositions and chemical processes, as we show here with WASP-39b. Using our model grids, we find that WASP-39b's best-fitting atmospheric metallicity is roughly ten times solar. In the limit of equilibrium chemistry, our non-detection of $CH_4$ at 3.4 μm paired with the prominence of the large $CO_2$ feature at 4.4 μm are indicative of a

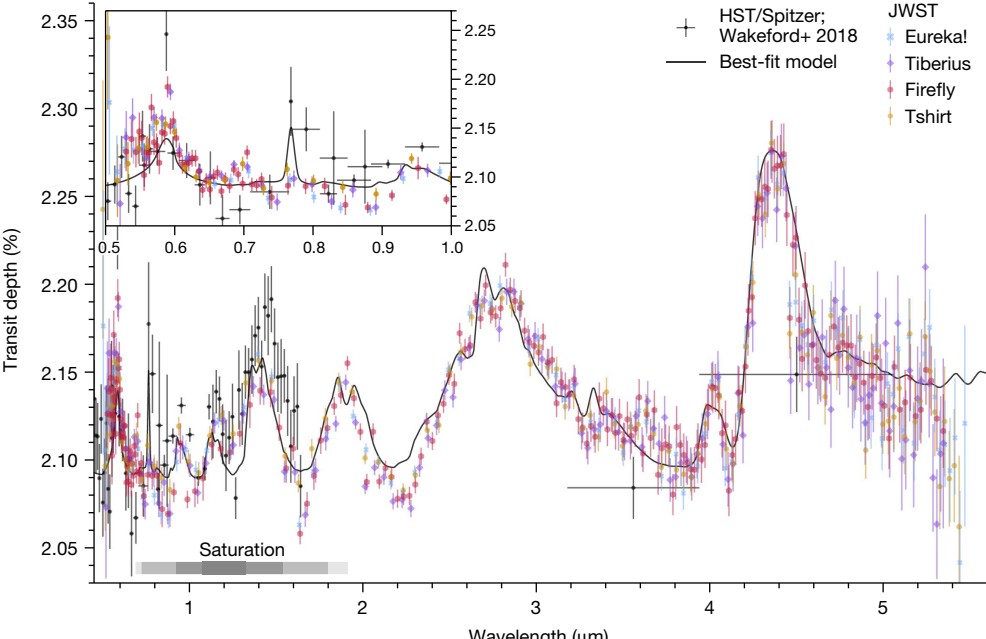

**Fig. 3 | WASP-39b transmission spectral measurements.** A comparison of the JWST transmission spectra obtained from the four independent reductions considered in this work (coloured points), which are all in broad agreement. Previous measurements from HST, VLT and Spitzer[6] are also shown (grey) along with our fiducial best-fit spectrum model from the PICASO 3.0 grid (black line).

All the transmission spectral data have $1-\sigma$ error bars shown. The saturated region of the detector is indicated (grey bar) with the shading representative of the level of saturation (also Extended Data Fig. 6). Different reductions are presented on slightly different wavelength grids for visual purposes, the original resolution each reduction used is discussed in the Methods.

super-solar atmospheric metallicity, as illustrated in Extended Data Fig. 9. This may point to WASP-39b's puffy envelope bearing more compositional similarity to the similarly massed ice giants than the gas giants. Moreover, the probably detection of $SO_2$, and its unexpectedly high estimated abundance, suggests that photochemical processes are pushing this species out of equilibrium. Photochemistry models show that sulfur compounds such as $H_2S$ efficiently photodissociate and recombine to form $SO_2$ with roughly 1 ppm abundances and at

$1$–$100$ mbar pressures[26]—roughly the same pressure range probed by our transmission spectroscopy (Extended Data Fig. 10). The abundance measurement of $SO_2$ can therefore serve as an important tracer of the thermochemical properties of highly irradiated stratospheres and the efficiency of photochemistry. Furthermore, our detection of a qualitatively significant wavelength dependence to the planet's central transit time (Extended Data Fig. 3) suggests that these observations are sensitive to differences in the atmospheric composition at

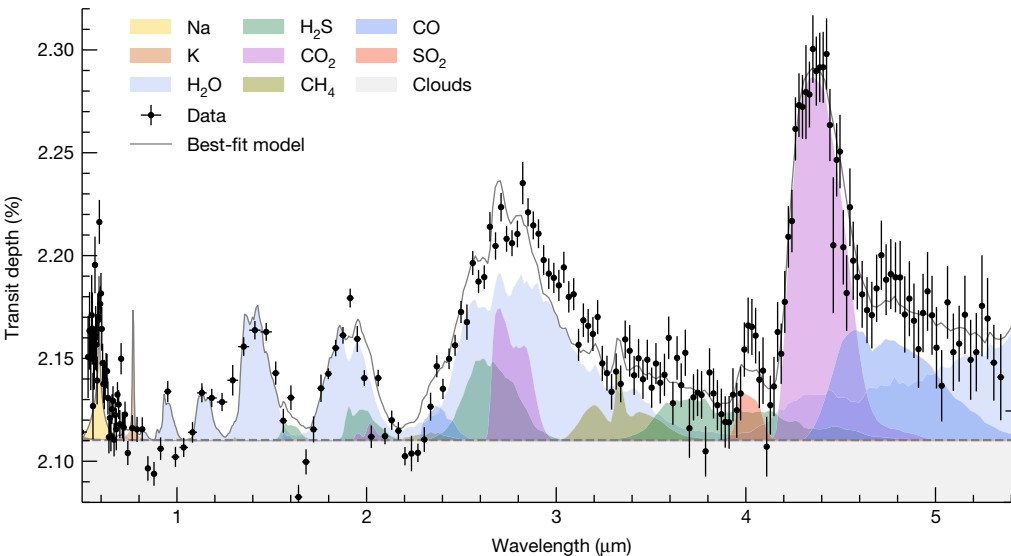

**Fig. 4 | The JWST-PRISM transmission spectrum of WASP-39b with key contributions to the atmospheric spectrum.** The black points with error bars correspond to the measured FIREFLY transit depths of the spectrophotometric light curves at different wavelengths. The best-fitting model spectrum from the PICASO 3.0 grid is shown as the grey line and the coloured regions correspond to the chemical opacity contributions at specific wavelengths. The best-fitting 1D RCTE model corresponds to a super-solar metallicity and super-solar

carbon-to-oxygen ratio with moderate cloud opacity (Methods). The PRISM transmission spectrum is explained by contributions from Na ($19\sigma$), $H_2O$ ($33\sigma$), $CO_2$ ($28\sigma$), CO ($7\sigma$), $SO_2$ ($2.7\sigma$) and clouds ($21\sigma$). The data do not provide evidence of $CH_4$, $H_2S$ and K absorption (Methods). Also, note that the detector was saturated to varying degrees between 0.8 and 1.9 μm. As before, the error bars are $1-\sigma$ standard deviations.

the planet's leading and trailing hemispheres. The measured roughly 20 s amplitude of this effect is in line with model expectations[30]. This indicates that such observations will be informative in exploring the three-dimensional (3D) nature of hot Jupiter atmospheres, which may give a more holistic understanding of their heat redistribution and nightside chemistry.

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

[1]Department of Earth and Planetary Sciences, Johns Hopkins University, Baltimore, MD, USA. [2]Department of Physics and Astronomy, Johns Hopkins University, Baltimore, MD, USA. [3]Department of Astronomy and Astrophysics, University of California, Santa Cruz, Santa Cruz, CA, USA. [4]Johns Hopkins APL, Laurel, MD, USA. [5]Center for Astrophysics, Harvard and Smithsonian, Cambridge, MA, USA. [6]Department of Physics, Imperial College London, London, UK. [7]Steward Observatory, University of Arizona, Tucson, AZ, USA. [8]School of Earth and Space Exploration, Arizona State University, Tempe, AZ, USA. [9]Institute of Research on Exoplanets, Department of Physics, University of Montreal, Montreal, Québec, Canada. [10]NASA Ames Research Center, Moffett Field, CA, USA. [11]School of Earth and Planetary Sciences, National Institute of Science Education and Research (NISER), HBNI, Jatani, India. [12]Department of Physics, Utah Valley University, Orem, UT, USA. [13]Department of Astronomy, University of Michigan, Ann Arbor, MI, USA. [14]Department of Astronomy and Carl Sagan Institute, Cornell University, Ithaca, NY, USA. [15]Lunar and Planetary Laboratory, University of Arizona, Tucson, AZ, USA. [16]School of Physics, University of Bristol, HH Wills Physics Laboratory, Bristol, UK. [17]Space Telescope Science Institute, Baltimore, MD, USA. [18]Department of Astronomy and Astrophysics, University of Chicago, Chicago, IL, USA. [19]Department of Astrophysical and Planetary Sciences, University of Colorado, Boulder, CO, USA. [20]Department of Physics and Astronomy, University of Kansas, Lawrence, KS, USA. [21]Earth and Planets Laboratory, Carnegie Institution of Washington, Washington, DC, USA. [22]Max Planck Institute for Astronomy, Heidelberg, Germany. [23]Harvard and Smithsonian, Center for Astrophysics, Cambridge, MA, USA. [24]INAF - Astrophysics Observatory at Turin, Turin, Italy. [25]School of Physics, Trinity College Dublin, Dublin, Ireland. [26]Laboratoire d'Astrophysique de Bordeaux, CNRS, Université de Bordeaux, Pessac, France. [27]University Observatory Munich, Ludwig Maximilian University, Munich, Germany. [28]Exzellenzcluster Origins, Garching, Germany. [29]Université Côte d'Azur, Observatoire de la Côte d'Azur, CNRS, Laboratoire Lagrange, Nice, France. [30]Atmospheric, Oceanic and Planetary Physics, Department of Physics, University of Oxford, Oxford, UK. [31]Department of Physics, University of Oxford, Oxford, UK. [32]Centre for Exoplanets and Habitability, University of Warwick, Coventry, UK. [33]Department of Physics, University of Warwick, Coventry, UK. [34]Indian Institute of Technology, Indore, Indore, India. [35]School of Physical Sciences, The Open University, Milton Keynes, UK. [36]Anton Pannekoek Institute for Astronomy, University of Amsterdam, Amsterdam, the Netherlands. [37]Centre for Astrobiology (CSIC-INTA), European Space Astronomy Centre Campus, University of Maria de Maeztu, Madrid, Spain. [38]BAER Institute, NASA Ames Research Center, Moffet Field, Mountain View, CA, USA. [39]New York University Abu Dhabi, Abu Dhabi, United Arab Emirates. [40]Center for Astro, Particle and Planetary Physics (CAP3), New York University Abu Dhabi, Abu Dhabi, UAE. [41]School of Physics and Astronomy, University of Leicester, Leicester, UK. [42]European Space Agency (ESA), ESA Baltimore Office, Baltimore, MD, USA. [43]Department of Physics and Astronomy, University College London, London, UK. [44]Centre for Exoplanet Science, University of St Andrews, St Andrews, UK. [45]Leiden Observatory, Leiden University, Leiden, the Netherlands. [46]Department of Astrophysical Sciences, Princeton University, Princeton, NJ, USA. [47]Department of Physics and Astronomy, KU Leuven, Leuven, Belgium. [48]Planetary Science Group, Department of Physics and Florida Space Institute, University of Central Florida, Orlando, FL, USA. [49]Jet Propulsion Laboratory, California Institute of Technology, Pasadena, CA, USA. [50]Division of Geological and Planetary Sciences, California Institute of Technology, Pasadena, CA, USA. [51]Institute for Astrophysics, University of Vienna, Vienna, Austria. [52]Department of Astronomy, University of Maryland, College Park, MD, USA. [53]California Institute of Technology, IPAC, Pasadena, CA, USA. [54]Department of Astronomy, University of Geneva, Geneva, Switzerland. [55]Department of Physics, University of Rome 'Tor Vergata', Rome, Italy. [56]Department of Physics and Astronomy, Faculty of Environment, Science and Economy, University of Exeter, Exeter, UK. [57]SRON Netherlands Institute for Space Research, Leiden, the Netherlands. [58]Institute for Astrophysics of Canarias (IAC), La Laguna, Tenerife, Spain. [59]Department of Astrophysics, University of La Laguna, La Laguna, Tenerife, Spain. [60]INAF Äi Palermo Astronomical Observatory, Palermo, Italy. [61]Department of Earth, Atmospheric and Planetary Sciences, Massachusetts Institute of Technology, Cambridge, MA, USA. [62]Kavli Institute for Astrophysics and Space Research, Massachusetts Institute of Technology, Cambridge, MA, USA. [63]Astronomy Department and Van Vleck Observatory, Wesleyan University, Middletown, CT, USA. [64]Institute of Astronomy, University of Cambridge, Cambridgeshire, UK. [65]Astrophysics Group, Keele University, Staffordshire, UK. [66]UVSQ, CNRS, CEA, Maison de la Simulation, Université Paris-Saclay, Gif-sur-Yvette, France. [67]Department of Physics, Brown University, Providence, RI, USA. [68]Université de Paris Cité and Univ Paris Est Creteil, CNRS, LISA, Paris, France. [69]Astrophysics and Planetary Sciences, University of Colorado Boulder, Boulder, CO, USA. [70]Department of Earth and Planetary Sciences, University of California Santa Cruz, Santa Cruz, CA, USA. [✉]e-mail: zafar@jhu.edu

## Methods

### Data reduction

One transit of WASP-39b was observed with the NIRSpec PRISM mode, with the 8.23-h observation roughly centred around the transit event. We used NIRSpec's Bright Object Time Series (BOTS) mode with the NRSRAPID readout pattern, the S1600A1 slit (1.6" × 1.6") and the SUB512 subarray. Throughout the exposure, we recorded 21,500 integrations, each with five 0.28-s groups up the ramp. We achieved a duty cycle of 82%.

We extracted transmission spectra of WASP-39b using four different reductions with the FIREFLy, tshirt, Eureka!+ExoTEP and Tiberius pipelines. The results from all reductions are broadly consistent (Fig. 3 and Extended Data Fig. 1). We used the FIREFLy reduction as our baseline for comparison to models throughout this paper, however, equivalent overall results can be deduced from the other reductions. Some key attributes of the reductions are compared in Extended Data Table 2. All reductions correct for 1/$f$ noise: correlated frequency-dependent read noise in the images caused by detector readout and current biases in the electronics[31]. We note that as the GAINSCALE step of the JWST pipeline applies a gain correction to the raw count rate files, the counts and count rates quoted herein are in units of electrons and electrons per second, respectively.

We find that recovery of the saturated region was possible by applying several custom steps described here. Without these steps, the heavily saturated region showed a large and unexpected point-to-point scatter of several thousand ppm in the transmission spectra. We note that there was limited on-sky NIRSpec calibration data available when the data were obtained and reduced, including an incomplete detector bias image whose values were all set to zero. We used a custom bias frame for this step (private communication, S. Birkmann). Although the transmission spectra longwards of about 2 μm could be extracted without the use of this calibration, we found that bias correction was critical to extract the spectrum in the saturated region.

In addition, to recover the saturated region it was necessary to perform a reference pixel correction something that was skipped by the default jwst pipeline for NIRSpec PRISM because no official reference pixels are present in the subarray (tshirt section below). All reductions also expand the saturation flags along entire columns and only use the groups before saturation for slope fitting in these regions. With these steps, the spectra broadly matched previous HST and Very Large Telescope (VLT) observations[6], with improvement in the region with only one or two groups before saturation. We expect that as updated NIRSpec calibration data become available, the recovery of saturated regions in PRISM observations may become easier, however, we still suggest avoiding rapid saturation with less than two groups before saturation if possible, especially if that region of the spectrum is important to one's science case.

**FIREFLy.** We performed custom calibrations on the uncalibrated data, including 1/$f$ noise destriping[9] at the group level, bad and hot pixel cleaning, cosmic ray removal and 5$\sigma$ outlier rejection. Destriping the data also removed potential background in the 2D images, although none was apparent in the data. The jump-step and dark-current stages of the jwst pipeline[12] (v.1.6.2) were skipped, and the top and bottom six pixels of the non-illuminated subarray were manually set to be reference pixels in the jwst pipeline reference pixel step. To obtain our final wavelength calibration, we extrapolated the STScI-provided in-flight instrumental wavelength calibration data product across the detector edge pixels that did not have an assigned wavelength. The calibration was derived using the ground-based wavelength solution. We performed tests to search for zero-point offsets in the calibration versus the planetary and stellar spectra and did not find any at the level of half a pixel width or greater.

JWST detectors integrate using a non-destructive up-the-ramp sampling technique in which the flux is measured in counts per second from fitting the ramp from the groups contained within each integration. Extended Data Fig. 2 shows the regions of the spectrum affected by saturation. Within a column where a pixel was marked as saturated by the pipeline in any given group, we used only the data from the preceding groups for ramp fitting and manually set an entire column of the detector as saturated if a pixel in that column was saturated. Because a small portion of the spectrum reaches our saturation threshold in the second group, this region of the spectrum only uses one group to derive a 'ramp'. Although we were able to recover the spectra in this wavelength range by flagging and ignoring saturated pixels at the group level, we note that the data quality is lower in the saturated region than in the rest of the spectrum given the counts per second ramp was measured from fewer than the total five groups.

We measured the positional shift of the spectral trace across the detector throughout the time series using cross correlation and used them to shift-stabilize the images with flux-conserving interpolation. This procedure reduced the amplitude of position-dependent trends in the light curves. We optimized the width of our flux extraction aperture at each wavelength pixel and extracted the spectrophotometry. For each wavelength, we tested a wide range of aperture widths and determined the width that minimized the scatter of the photometry of the first 350 data points. We bin the cleaned spectrophotometry in wavelength to create 207 variable-width spectral channels with roughly $10^5$ counts per second in each bin and widths ranging from 3.3 to 60 nm. Because we use fewer groups in the saturated detector columns, our bin widths are larger by a factor of a few in this region to account for the lower count rates per detector column.

Before fitting the transmission spectrum, we use a very wide, high-SNR white light channel (3–5.5 μm) to fit for the planet's orbital parameters (listed in Extended Data Table 1). Restricting the wide bin to the reddest wavelengths minimizes the impact of limb darkening on the transit light curve and the resulting covariance with the orbital system parameters while ignoring the saturated region. We fit this white light curve using the Markov chain Monte Carlo sampler emcee[32] within the least-squares minimization framework of lmfit. We use 1,000 steps and uniform priors with extremely wide bounds that encapsulate the limits of physicality to ensure that there is no bias introduced by the prior. Our fitting approach accounts for non-Gaussian degeneracies in the posterior distribution, thereby addressing the known linear correlation between impact parameter ($b$) and the scaled semimajor axis ($a/R_*$)

We excluded the first 3,000 integrations as they showed a slight non-linear baseline flux trend, and integrations 20,750–20,758 because of a high-gain antenna move that was identified from outliers in the photometry that correlated with noticeable trace shifts in the $x$ and $y$ directions. To measure the transmission spectrum, we fit the light curve at each wavelength channel jointly with a transit model[33] and a linear combination of systematics vectors composed of the measured spectral shifts in the $x$ and $y$ directions. At each channel, we fit the planet's transit depth and the stellar limb darkening, while fixing the transit centre time $T_0$, affect parameter $b$ and normalized semimajor axis $a/R_*$ to the values determined in the white light curve fit. We also fix the orbital period to the published value of 4.0552941 days (ref. [34]). With the orbital system parameters fixed, we find the posterior distribution is well-fit by a multivariate Gaussian distribution, and therefore use a Levenberg–Marquart least-squares minimization algorithm[24] to efficiently determine the best-fit parameters. In each channel, we inflate the transit depth error bars in quadrature with the measured residual red noise in the photometry as measured by the binning technique[35]. Measured uncertainties on the transit depths vary from 50 to 200 ppm, with a median of 99 ppm (Extended Data Fig. 4). As the noise levels are very close to the limit with what is expected including only photon and read noise sources, tools such as PandExo[36] should accurately predict what is achievable for other planets. We measure an increase in red noise for a few select spectral channels, but otherwise the light curves show no significant systematic errors, with some channels binning down to

precision levels of a few ppm. We measure $x$ and $y$ jitter systematics at the roughly 100 ppm level. We see differences in the central transit time as a function of wavelength on the order of 10 s, which may be attributable to limb asymmetries in the atmospheric temperature and composition. We show these signatures in Extended Data Fig. 3. Notably, we see a significant timing structure in the 2–3 µm range, which may arise from limb asymmetries in temperature and/or cloud coverage at the altitude probed by the water vapour absorption feature at 2.7 µm (ref. [37]). Further analysis of the spectrophotometry could be warranted to investigate limb asymmetries in more detail.

We fit the transit light curves using a quadratic function to model stellar limb darkening given as,

$$\frac{I(\mu)}{I(1)} = 1 - a(1-\mu) - b(1-\mu)^2 \qquad (1)$$

where $I(1)$ is the intensity at the centre of the stellar disc, $\mu = \cos(\theta)$ where $\theta$ is the angle between the line of sight and the emergent intensity, and $a$ and $b$ are the limb darkening coefficients. We tested a four-parameter non-linear limb darkening function[38] as well, which provided equivalent results. In practice, we first fit for both $u_+ = a + b$ and $u_- = a - b$ for the quadratic law. When comparing the limb darkening coefficients to theoretical values, we find an offset between the theoretically derived values of $u+$ from the 3D stellar models from ref. [39] and the JWST values derived from the transit light curve fits (Extended Data Fig. 5). This offset suggests the limb of WASP-39A is brighter than the stellar models predict. We fit for this offset and find it to be $-0.065 \pm 0.022$. As the wavelength-to-wavelength shape of $u_+$ is well described by the model, we then apply this offset to the theoretical limb darkening coefficients and then subsequently fix $u_+$ while allowing only $u_-$ to be free (Extended Data Fig. 5). This procedure helps reduce degeneracies when fitting several limb darkening coefficients and increases the precision of the transmission spectrum, as the limb darkening is often not well constrained, particularly at long wavelengths where the limb darkening is weak[39] (Extended Data Fig. 5). The main effect of fitting for limb darkening over fixing the coefficients to the 3D models is the transit depth level of the optical spectrum, which is lower with values fixed to the model. We compare the optical spectrum with fixed limb darkening to the HST data from ref. [6] in Extended Data Fig. 6, which was also fit with limb darkening fixed to the same model. Overall, we find good agreement between the two spectra. We note that the assumptions around limb darkening can affect the optical spectra continuum wthat particularly affects the interpreted levels of aerosol scattering: further investigations are warranted.

**tshirt.** We use the tshirt pipeline, for example, ref. [40], to extract an independent set of light curves and spectrum. We begin with the uncalibrated 'uncal' data product and apply a custom set of processing steps on stage 1 that build on the existing jwst stage 1 pipeline software v.1.6.0 with reference files CRDS (Calibration Reference Data System) jwst_0930.pmap. We use a custom bias file shared by the instrument team (S. Birkmann, private communication), which is the same file that was delivered to the JWST CRDS.

We attempt to minimize the biasing effects of count rate non-linearity by modifying the quality flags of pixels surpassing 90% of full-well depth at the group stage. To ensure that there are no systematic differences between pixels within the spectral trace and in the background region, we adjust the quality flags uniformly along the entire pixel column at each group for all integrations. We skip the 'jump' and 'dark' steps of stage 1.

The tshirt code includes a row, odd-even by amplifier correction to reduce $1/f$ noise. We first identify source pixels by choosing pixels with more than five data numbers per second (DN s$^{-1}$) in the rate file and expanding this region out by 8 pixels. We then identify background pixels for $1/f$ corrections by choosing all non-source pixels and pipeline flagged non-'DO NOT USE' pixels. We loop through every group and subtract the median of odd (even) row background pixels from all odd (even) rows. We next find a column-by-column median of all background pixels to calculate a $1/f$ stripe correction and subtract this from each column.

After calculating rate files in DN s$^{-1}$, we use tshirt to perform covariance-weight extraction of the spectrum[31]. We do a column-by-column linear background subtraction using pixels 0–7 and 25–32. We use a rectangular source extraction region centred on $Y = 16$ pixels with a width of 14 pixels. We assume the correlation between pixels to be 8% from previous studies of background pixels[31]. We use a spline with 30 knots to estimate a smooth spectrum of the star at the source pixels and identify bad pixels as ones that deviate by more than $50\sigma$ from the spline. Pixels that are more than $50\sigma$ or else marked as DO NOT USE are flagged and then the spatial profile is interpolated over those pixels. No corrections were made to the centroid or wavelength solution because of the exceptional pointing stability of the observatory[41].

When fitting the light curves, we exclude all time samples between UT 2022-07-10T23:20:01 and 2022-07-10T23:21:08 to avoid the effects of the high-gain antenna move. We first fit the broadband light curve with all wavelengths. We assume zero eccentricity and the orbital parameters from ref. [34] for $a/R_*$ and period. We try fitting the white light curve with eccentricity and argument of periastron set free and find that eccentricity is consistent with 0. We therefore assume zero eccentricity and a transit centre projected to the time of observations from a fit to the TESS data. We also assume an exponential temporal baseline in time to the data and a second-order polynomial trend in time. We fit the quadratic limb darkening parameters with uninformative priors[42] and the exoplanet code[43–45] with 3,000 burn-in steps and 3,000 sampling steps and two No U Turns Sampling chains[46]. We next binned the spectra into 116 bins, each 4 pixels wide. We fit all the individual spectroscopic channels with the orbital parameter fixed from the broadband light curve fit and only allowed the transit depth and limb darkening parameters to be free. Our resulting transit depth uncertainties ranged from 35 to 732 ppm, with a median of 90 ppm.

**Eureka! and ExoTEP.** We use the Eureka! pipeline[47] for the data reduction steps of detector processing, data calibration and stellar spectrum extraction, and the ExoTEP pipeline[48–50] to generate light curves in each wavelength bin and perform light curve fitting.

We start our data reduction using the uncalibrated uncal outputs of the jwst pipeline's stage 0. From there, Eureka! acts as a wrapper for the first two stages of the jwst pipeline, v.1.6.0. We use the jwst pipeline to fit slopes to the ramp in each pixel and perform data calibration, and follow the default pipeline steps unless otherwise stated. We skip the jump detection step, meant to correct the ramps for discontinuities in the slopes of group count rates as a function of time. Owing to the small number of groups up the ramp, performing this step leads to a large fraction of the detector pixels being incorrectly flagged as outliers and we therefore rely on the time series outlier-clipping steps in the subsequent stages to correct for cosmic rays. A custom bias frame is used, rather than the default one available on CRDS at the time of reduction. We also expand the saturation flags in stage 1 to ignore saturated pixels more conservatively than allowed by the default jwst pipeline settings: in each group, we flag pixels as saturated if they reach roughly 85% of the full well in the median image across all integrations for that group and expand the saturation flag such that in a given detector column (constant wavelength) all pixels are marked as saturated if any one pixel in that column is flagged. This is implemented by inputting the indices of columns to mask on the basis of an inspection of the uncal data products, rather than an internal calculation of the full well percentage. We include a version of the row, odd-even by amplifier correction described above, using the top and bottom six rows. We further add a custom background correction at the group level before ramp fitting, and subtract from each column the median of

the six pixels at the top and at the bottom of the detector, excluding outliers at more than the 3-$\sigma$ level. We skip the 'photom' step in stage 2 of the STScI detector pipeline because absolute fluxes are not needed in our analysis. We also skip the 'extract1d' step as we perform custom spectral extraction using Eureka!.

For 1D spectral extraction, we trim the array to include only columns 14 to 495 in the dispersion direction, as NIRSpec's throughput is negligible beyond this range. We then use the median detector frame to construct the weights used in the optimal extraction based on ref. [51]. Pixels are masked if they have an marked data quality flag (that is, bad pixels that are flagged by the jwst pipeline as 'DO NOT USE' for various reasons) or if they are clipped by two iterations of 10-$\sigma$-clipping of the time series. We perform the optimal extraction over eight rows centred on the source position (corresponding to a spectral half-width aperture of 4 pixels). The source position is identified from the maximum of a Gaussian fitted to the summed spatial profile from all detector columns over the entire integration.

We use ExoTEP to generate median-normalized light curves at the native pixel resolution from each detector column, using the stellar spectra outputs from stage 3 of Eureka!. We then perform further clipping of outliers in time in the white and wavelength-dependent light curves by computing a running median with a window size of 20 and excluding 3$\sigma$ outliers in several time series. This outlier-clipping was applied to the flux, source position and width in the cross-dispersion direction in each frame and spectrum shifts in the dispersion direction.

We jointly fit astrophysical and systematics model parameters to the white (0.5–5.5 μm) light curves and each of the wavelength-dependent light curves. Our astrophysical transit model is calculated using the batman package[33]. Using the white light curve, we fit for the two coefficients of a quadratic limb darkening law (equation (1)), WASP-39b's impact parameter, scaled semimajor axis $a/R_*$, time of transit centre and the planet-to-star radius ratio. In each of the wavelength channels we then fix the planet's impact parameter, semimajor axis and transit time to the values derived from the white light curve and fit only for the planet-to-star radius ratio and the two quadratic limb darkening coefficients. For the systematics model, we assume a linear trend with time that can be different in each spectroscopic channel, and fit for its slope and $y$ intercept. Last, we fit a single-point scatter to each light curve, which illustrates the level of scatter required for our joint model to reach a reduced chi-squared of 1. The fitted light curve scatter in both the white light curve and wavelength-dependent channels is within a few percent of the expectation from the high-frequency scatter in the raw light curves, which attests to the lack of systematics. We bin the final transmission spectrum (four points binned together throughout the spectrum) for visual comparison with the other reductions in Fig. 3.

**Tiberius.** The Tiberius pipeline builds on the LRG-BEASTS spectral reduction and analysis pipelines introduced in refs. [16,52,53]. The Tiberius pipeline operates on the stage 1 JWST data products to obtain 1D stellar spectra through tracing of the stellar spectra, fitting and removal of the background noise and simple aperture photometry. We used the FIREFLy-processed stage 0 data.

Before tracing the spectra, we interpolate each column of the detector onto a finer grid, 10× the initial spatial resolution. This step improves the extraction of flux at the subpixel level, particularly where the edges of the photometric aperture bisect a pixel, and leads to a 14% reduction in the noise in the data. We also interpolate over the bad pixels using their nearest neighbouring pixels in $x$ and $y$. We identify bad pixels by combining 5$\sigma$ outlying pixels found by means of running medians operating along the pixel rows with bad pixels identified by visual inspection. We trace the spectrum by fitting a Gaussian distribution at each column (in which a column refers to the cross-dispersion direction) to the stellar spectra. We then use a running median, calculated with a moving box with a width of five data points, to smooth the measured centres of the trace. We fit these smoothed centres with a fourth-order polynomial, removed five median absolute deviation outliers and refitted with a fourth-order polynomial.

To remove residual background flux not captured by the 1/$f$ correction, we fit a linear polynomial along each column in the spatial direction. We mask the stellar spectrum, defined by an aperture with a full width of 4 pixels centred on the trace we found in the previous step, from this background fit. We also mask an extra 7 pixels on either side of this aperture so that the background fit is not affected by the wings of the stellar point spread function. This left us with 7 pixels at each edge of the detector (a total of 14 pixels) with which to estimate the background. We also clipped any pixels within the background that deviate by more than three standard deviations from the mean for that particular column and frame to avoid residual bad pixels and cosmic rays affecting our background estimation. We found that this extra background step led to a 3% improvement in the precision of the transmission spectrum.

The stellar spectra are then extracted by summing the flux within a 4-pixel-wide aperture following the removal of the background at each column. The background count level, as estimated by the JWST Exposure Time Calculator is on the order of a few counts per second, meaning the background is negligible. Further, because we perform 1/$f$ subtraction, this faint background is subtracted column-by-column. We experimented with the choice of the aperture width, also running reductions with 8- and 16-pixel-wide apertures. The 8-pixel-wide aperture gave a median uncertainty 1% larger than a 4-pixel aperture and a 16-pixel aperture gave an uncertainty 15% larger than 4 pixels. This same change was reflected in the median root mean square of the residuals to the light curve fits. As the stellar point spread function is so narrow in PRISM data, we believe that the increase in noise with increasing aperture width is related to the increasing influence of photon noise, read noise and bad pixels where the stellar flux is lower. Following the extraction of the stellar spectra, we divide the measured count rates by a factor of ten to correct for our pixel oversampling, as described above.

To remove residual cosmic rays, we identify outliers in each stellar spectrum through comparison with the median stellar spectrum. We did this in three iterations, each of which involves making a median spectrum, identifying outliers (10, 9, 8 $\sigma$) and replacing pixels containing a cosmic ray with a linear interpolation between neighbouring pixels. We tested this interpolation against assigning the cosmic ray pixels zero weight and found that this led to a negligible difference in the transmission spectrum. To correct for shifts in the stellar spectra and align each spectrum in pixel space, we cross-correlate each stellar spectrum with the first spectrum of the observation and linearly resample each spectrum onto a common wavelength grid. We adopt the custom wavelength solution calculated by the tshirt pipeline, which uses the jwst pipeline to evaluate the wavelengths at pixel row 16 using the world coordinate system.

Our white light curves are created by summing over the full wavelength range between 0.518 and 5.348 μm. We make two sets of spectroscopic light curves: one set of 440 light curves at 1-pixel resolution and one set of 147 light curves at 3-pixel resolution. We mask integrations 20,751–20,765 because of a high-gain antenna move that leads to increased noise in the light curves. We also mask the first 2,000 integrations from our analysis because of a systematic ramp. This means our light curves each contained 19,486 data points.

To fit our light curves, we began by fitting the white light curve to determine the system parameters.

We fit for the following parameters: the scaled planetary radius ($R_p/R_*$), the planet's orbital inclination ($i$), the time of mid-transit ($T_C$), the scaled separation ($a/R_*$), the linear limb darkening coefficient ($u_1$) and the parameters defining the systematics model. We fix the planet's orbital period to 4.0552941d and eccentricity to 0 (ref. [34]). For the remaining parameters, we use the values from ref. [34] as initial guesses.

For the analytic transit light curve model, we use batman[33] with a quadratic limb darkening law. We use ExoTiC-LD[54,55], with 3D stellar models[39] to determine the appropriate limb darkening coefficients (LDs), adopting the stellar parameters ($T_{eff}$ = 5,512 ± 55 K, log $g$ = 4.47 ± 0.03 cgs, [Fe/H] = 0.01 ± 0.09 dex) from ref. [34] and Gaia DR3 (refs. [56,57]). For our final fits, we fix the quadratic coefficient, $u_2$, to the values determined by ExoTiC-LD. However, we also run a set of fits with neither $u_1$ nor $u_2$ fixed and find this leads to a transmission spectrum that is qualitatively similar to the one in which LDs are fixed. For the systematics model, we sum the following three polynomials: quadratic in time, linear in $x$ position of the star on the detector and linear in $y$ position of the star on the detector. The final fit model, $M$, was of the form:

$$M(t) = T(t, p) \times \left( \sum_i (S_i(a_i, s)^{n_i}) \right) \quad (2)$$

where $t$ is time, $p$ are the parameters of the transit model, $T$, $a$ are the ancillary data and $s$ are the parameters (polynomial coefficients) of the systematics model, $S$. The systematics model is the sum of the polynomials operating over each ancillary input, $a_i$, with $n_i$ defining the order of the polynomial used for each input.

We fit our white light curve in three steps: a first fit to remove any $4\sigma$ outliers from the light curves, a second fit that is used to rescale the photometric uncertainties such that the best-fitting model gives $\chi_v^2 = 1$ and a third fit with the rescaled photometric uncertainties, from which our final parameter values and uncertainties are estimated. The parameter uncertainties were calculated as the standard deviation of the diagonal of the covariance matrix that was in turn calculated from the Jacobian returned by scipy.optimize.

Following the white light curve, we fit our spectroscopic, wavelength-binned, light curves. For these fits, we held $a/R_*$, $i$ and $T_C$ fixed to the values determined from the white light curve fit: 11.462 ± 0.014, 87.847 ± 0.015°, 2,459,770.835623 ± 0.000008 Barycentric Julian Date Dynamical Time ($BJD_{TDB}$). These values are different from the FIREFLy-reduced white light parameters, and these differences will be explored in greater detail in a future work. To zeroth order, offsets in orbital parameters result in simple vertical offsets in the resulting transmission spectrum. The remaining fit parameters were the same as for the white light curve fit. We perform the same iteration of fits using a Levenberg–Marquardt algorithm to determine $R_p/R_s$ as a function of wavelength.

### Reduction comparison
Procedural differences exist across the four main reductions of the dataset, which may account for the subtle qualitative differences between the final reduced spectra. A careful investigation of these nuances is warranted and will be presented in a future paper. Extended Data Table 2 highlights some key procedural differences between the reductions. We note that, despite these differences, the resulting exoplanet spectra are qualitatively in excellent agreement with each other (Fig. 3) because of the stability of the data and the self-calibrating nature of the transit technique.

### Stellar activity
WASP-39b has a reported low activity level[8], with a Ca II H and K stellar activity index of $\log R'_{HK}$ = −4.994 (ref. [4]). NGTS and TESS photometric monitoring of WASP-39A is reported in ref. [22], which finds low modulations at the 0.06% level with no apparent star-spot crossings. With low stellar activity levels, the transit observations are unlikely to be affected by stellar activity.

### Forward model grids
We use four different 1D RCTE model grids to assess atmospheric properties such as detection of individual gases, metallicity, carbon-to-oxygen (C/O) elemental abundance ratio, and the presence/absence of clouds.

The ScCHIMERA[58,59], PICASO 3.0 (refs. [60–63]), ATMO[54,64,65] and PHOENIX[66,67] models were used to generate these grids specifically for WASP-39b. Whereas the ATMO and the PHOENIX grids were used to fit the data with a reduced $\chi^2$ based grid search method, the PICASO 3.0 and ScCHIMERA grids were used in a grid retrieval framework using a nested sampler[68,69]. Within each nested sample likelihood calculation, the transmission spectra are generated on-the-fly by postprocessing the precomputed 1D RCTE model atmospheres. The $SO_2$ volume mixing ratio and cloud properties are injected into spectrum during this postprocessed transmission calculation. Extended Data Fig. 7 shows best-fit models obtained by each of the four grids compared with the transmission spectrum obtained with the FIREFLy data reduction pipeline. ScCHIMERA, PICASO 3.0 and ATMO produce fits with reduced $\chi^2$ between 3.2 and 3.3, while the PHOENIX grid obtains a reduced $\chi^2$ of 4.3. The reduced $\chi^2$ is defined as the total $\chi^2$ calculated from all the data points divided by the total number of data points. Although PICASO 3.0, ScCHIMERA and ATMO predict the metallicity of the atmosphere to be about 10× solar, PHOENIX finds a best-fit metallicity to be a 100× solar that might be due to the larger grid spacing of the PHOENIX grid along both the cloud and metallicity dimensions. Although the models qualitatively match the data, the reduced $\chi^2$ obtained by the best-fitting models from these grids are also greater than three, which suggests that these are not fitting the data particularly well. These poor fits could arise for many reasons, such as the region of the data affected by saturation, the presence of disequilibrium chemistry in the atmosphere due to vertical mixing or photochemistry and the non-grey nature of scattering in the upper atmosphere. Extended Data Table 3 provides a summary of the best-fit atmospheric parameters obtained by the four different grids with different fitting methods (grid retrievals and grid search). To explore the effect of the saturated region on the best-fit parameters, we inflate the transit depth errors in the saturated regions (0.68–1.91 µm) by a factor of 1,000 and recompute the best-fit models using the grid retrieval framework with both the PICASO v.3.0 and ScCHIMERA grids. We find that this did not significantly change any of the best-fit parameters including the metallicity and the C/O ratio. Extended Data Table 3 lists the best-fit parameters obtained when the saturated region error bars were inflated by a factor of 1,000. We summarize the main results obtained by these 1D grids here and refer the reader to ref. [29] for detailed descriptions of each of these model grids.

### Detection significance of gases
We quantify the detection significance of each species through a Bayes factor analysis, for example ref. [70]. To do so within the ScCHIMERA grid retrieval framework, we remove each gas during the transmission spectrum computation step (the 1D RCTE atmosphere models remain unchanged) one at a time and re-run the nested sampler. We compare the Bayesian evidence of each removed-gas run to that of grid retrieval with all the gases. There is no change in the number of parameters except the cloud and $SO_2$ mixing ratio parameters. Extended Data Table 4 shows the result of this exercise summarized as the log-Bayes factor and a conversion to the detection significance: for example ref. [71].

We also quantify the detection significances of different gases following the procedure used in ref. [29]. To calculate the detection significance of each gas, the best-fit transmission spectrum model from the PICASO 3.0 grid ([M/H] = +1.0, C/O = 0.68) is recalculated without that gas. The wavelength ranges in which the particular gas has the most prominent effect are first identified and then a residual spectrum is calculated by subtracting the model without the gas from the data. The residual spectra for $H_2O$, $CO_2$, CO, Na, $SO_2$ and $CH_4$ are shown in the six panels of Extended Data Fig. 8. We fit each of these residual spectra with two functions, a Gaussian–double Gaussian–Voigt function and a constant line. We use the Dynesty nested sampling routine[68] to perform the fits and to determine the Bayesian evidence associated with each fit. The Bayes factor between the fits of the residual spectrum with the Gaussian–Voigt function and the constant line is then used to determine

the detection significance of a gas. For example, for computing the detection significance of $H_2O$, two adjacent $H_2O$ features between 1 and 2.2 μm are used. We note that $H_2O$ is expected to be the dominant opacity source in other wavelength ranges (for example, 2.2–3 μm) as well, so choosing two features for this analysis would produce a lower limit on the detection significance of $H_2O$. The best-fit double Gaussian function to these features along with its $1\sigma$ and $2\sigma$ envelopes are shown with the red line and shaded regions in Extended Data Fig. 8 top-left panel. The same residual spectrum is also fitted with a straight line shown with blue colour in Extended Data Fig. 8. The logarithm of the Bayes factor between the two models is found to be $\ln B = 242$, which shows that the model with $H_2O$ is significantly favoured over a model without any $H_2O$. The detection significance of $H_2O$ corresponding to this Bayes factor is calculated using the prescription in ref. [71] and is found to be $22\sigma$. The same methodology, but with a single Gaussian function, is also followed for $CO_2$, CO, $SO_2$, $H_2S$ and $CH_4$ to get their detection significance summarized in Extended Data Table 4, in the last column. Our Gaussian residual fit significance for $CO_2$ matches the initial analysis of the NIRSpec PRISM data presented in ref. [29].

As shown in Extended Data Table 4, the detection significance of all gases increases with the Bayes factor analysis technique relative to the Gaussian–Voigt function technique. This is notably also the case for $SO_2$, lending confidence to the detection and identification of the molecule, as the feature is better fit by its respective opacity profile.

### Resolution bias and the detection significance of CO
The resolution-linked bias effect serves to dilute the measured amplitudes of planetary atmospheric features because of overlapping absorption lines in the stellar atmosphere. Although this effect is negligible for most stars earlier than M dwarfs, some stellar CO absorption is expected in WASP-39, meaning the measured planetary CO abundance may be biased. Following equation 4 in ref. [72] and using high-resolution ($R$ of roughly $10^5$) PHOENIX models of the planet and the star, we quantify an upper limit on the magnitude of this bias effect. We find that the planetary CO feature is biased by 30 to 40 ppm in the 4.5–5.1 μm region, leading to as much as a roughly $1 - \sigma$ underestimate of the planetary CO absorption strength, and subsequently a similar underestimate of its abundance. We note that this effect is potentially weakened by Doppler broadening of the molecular lines (which is unaccounted for by PHOENIX) because of stellar rotation, planetary orbital radial velocity and planetary winds. Future work, which may benefit from more detailed modelling and high-resolution observations of WASP-39's CO band heads, will better quantify the magnitude of this dilution.

### Metallicity, C/O ratio and $CH_4$ abundance
The best-fitting atmospheric metallicity for WASP-39b is found to be roughly ten times the solar metallicity using the model grids. The top panel in Extended Data Fig. 9 shows the observed transmission spectrum of the planet between 2.0 and 5.3 μm (in which variations due to metallicity are most prominent), along with several transmission spectrum models assuming different atmospheric metallicities ranging from subsolar values (for example, 0.3× solar) to super-solar values (for example, 100× solar). The bottom panel demonstrates the effect of different atmospheric C/O ratios at ten times solar metallicity on many transmission spectrum models along with the data. As the star WASP-39 has near-solar elemental abundances[73], scaled solar abundances are a reasonable choice for this star. The $CH_4$ feature between 3.1–4 and 2.2–2.5 μm is very prominent in subsolar and solar metallicity thermochemical equilibrium models shown in Extended Data Fig. 9. The absence of such a $CH_4$ feature in the data is evident. This, combined with the large $CO_2$ feature between 4.3 and 4.6 μm and measurable CO feature at 4.7 μm, led to a super-solar (10×) metallicity estimate for the planet. The C/O ratio of the RCTE models significantly affects the predicted gas abundances, and therefore the calculated transmission spectrum. Extended Data Fig. 9 bottom panel shows that

for metal-rich atmospheres (for example, >10× solar) with C/O ratios lower than 0.7, the transmission spectrum is dominated by features of oxygen-bearing gases ($H_2O$, $CO_2$, CO): for example, refs. [65,74,75]. But for higher C/O ratios (for example, 0.916), the transmission spectrum becomes $CH_4$ dominated at wavelengths greater than 1.5 μm. We obtain an upper limit on the C/O ratio of WASP-39b at about 0.7. However, these interpretations are based on single-best fits from model grids assuming thermochemical equilibrium. Other chemical disequilibrium processes such as atmospheric mixing and high-energy stellar radiation-induced photochemistry can also potentially affect this interpretation. These disequilibrium chemistry effects require further exploration in the context of WASP-39b and will be discussed in future work (Welbanks et al. (in prep), Tsai et al. (submitted)).

The best-fitting metallicity models can be used to place an upper limit on the $CH_4$ abundance, if the pressure ranges probed by the transmission spectrum are estimated. To estimate the pressure ranges probed by the data, we use the best-fit PICASO 3.0 model to calculate a pressure- and wavelength-dependent transmission contribution function of the atmosphere[76]. This contribution function for the best-fit 10× solar metallicity PICASO v.3.0 model is shown as a heat map in Extended Data Fig. 10. This shows that the data mostly probes pressure ranges between 0.1 and 2 mbar. We also computed contribution functions for models with solar metallicity and find that they probe similar pressure ranges as well. Extended Data Fig. 10 also shows the pressure dependent $CH_4$ abundances in models with different metallicities presented in Extended Data Fig. 9 top panel. As only super-solar metallicity thermochemical equilibrium models are preferred by the data, the abundance profiles in Extended Data Fig. 10 help us in putting an upper limit of $5 \times 10^{-6}$ on the $CH_4$ volume mixing ratio between 0.1 and 2 mbar.

### Clouds
The observed spectrum shows slightly muted transit depths, across the entire wavelength range, compared with the depths expected from clear atmospheric models. This hints towards some extra opacity source in the atmosphere with weak wavelength dependence. Opacity sources such as clouds can mute the spectral features in a transmission spectrum[2,4]. We postprocess the transmission spectrum models with grey (that is, wavelength-independent) cloud opacities to check whether they are preferred over clear atmospheric models by the data. However, the treatment of clouds differ between the four 1D RCTE model grids. PICASO 3.0 and ScCHIMERA grids implemented the cloud opacities using the following equation,

$$\tau_{i,\text{cld}} = \kappa_{\text{cld}} \frac{\delta P_i}{g} \tag{3}$$

where $\tau_{i,\text{cld}}$ is the cloud optical depth of the $i$th atmospheric layer in the model with pressure width $\delta P_i$ and $g$ represents the gravity of the planet. The best-fit value of the grey cloud opacity $\kappa_{\text{cld}} = 10^{-2.07}$ cm$^2$ g$^{-1}$ is calculated in a Bayesian framework by postprocessing the RCTE model grid with this cloud opacity and comparing these postprocessed models with the data. The ATMO grid includes grey cloud decks at several pressures between 1 and 50 mbar, but with variable factors 0, 0.5, 1 and 5 governing cloud opacity with respect to $H_2$'s scattering cross-section at 0.35 μm, where a factor 0 indicates a cloud-free model spectrum. The PHOENIX grid includes similar cloud decks but between 0.3 and 10 mbar with cloud optical depth enhancement factors (identically defined as the ATMO grid) 0 and 10. We find that the cloudy models better fit the data than clear models across all four model grids. The contribution of clouds in limiting the depths of the gaseous features across the entire wavelength range is also shown in Fig. 4 with the grey shaded region.

### 4 μm $SO_2$ feature identification
None of the 1D RCTE models capture the 4 μm absorption feature seen in the data. We searched for several candidate gas species that could

produce this feature if their abundances differ from the expected abundances from thermochemical equilibrium. The list of searched chemical species include C-bearing gases such as $C_2H_2$, CS, $CS_2$, $C_2H_6$, $C_2H_4$, $CH_3$, CH, $C_2$, $CH_3Cl$, $CH_3F$, CN and CP. Various metal hydrides, bromides, flourides and chlorides such as LiH, AlH, FeH, CrH, BeH, TiH, CaH, HBr, LiCl, HCl, HF, AlCl, NaF and AlF were also searched as potential candidates to explain the feature. $SO_2$, $SO_3$, SO and SH are among the sulfur-based gases that were considered. Other species that were considered include gases such as $PH_3$, $H_2S$, HCN, $N_2O$, $GeH_4$, $SiH_4$, SiO, $AsH_3$, $H_2CO$, $H_3^+$, $OH^+$, KOH, $Br\alpha$-H, AlO, CN, CP, CaF, $H_2O_2$, $H_3O^+$, $HNO_3$, KF, MgO, PN, PO, PS, SiH, SiO2, SiS, TiO and VO.

Among all these gases, $SO_2$ was the most promising candidate in terms of its spectral shape and chemical plausibility, although the expected chemical equilibrium abundance of $SO_2$ is too low to produce the absorption signal seen in the data. However, previous work exploring photochemistry in exoplanetary atmospheres[25,26] have shown that higher amounts of $SO_2$ can be created in the upper atmospheres of irradiated planets through photochemical processes. Therefore, we postprocess the PICASO 3.0 and ScCHIMERA chemical equilibrium models with varying amounts of $SO_2$ in a Bayesian framework to estimate the $SO_2$ abundance required to explain the strength of the 4-μm feature. The required volume mixing ratio of $SO_2$ was found to be roughly $10^{-5}$–$10^{-6}$. Note that in obtaining this estimate we assumed that the $SO_2$ volume mixing ratio does not vary with pressure for simplicity. In a photochemical scenario this assumption is probably not realistic, although the pressure range probed by $SO_2$ is also limited. Whether photochemical models can produce this amount of $SO_2$ in the atmospheric conditions of WASP-39b is a pressing question that the ERS team is now exploring (Welbanks et al. (in prep), Tsai et al. (submitted)). Whether this feature can be better explained by any other gaseous absorber is also at present under investigation by the ERS team.

## Data availability

The data used in this paper are associated with JWST program ERS 1366 and are available from the Mikulski Archive for Space Telescopes (https://mast.stsci.edu). The data products required to generate Figs. 1, 2, 3, and Extended Data Figs. 1, 3 and 5 are available here: https://zenodo.org/record/7388032. All additional data are available upon request.

## Code availability

The codes used in this publication to extract, reduce, and analyse the data are as follows; STScI JWST Calibration pipeline (https://github.com/spacetelescope/jwst), FIREFLy[23], tshirt[40], Eureka![47] (https://eurekadocs.readthedocs.io/en/latest/) and Tiberius[16,52,53]. In addition, these made use of Exoplanet[43] (https://docs.exoplanet.codes/en/latest/), Pymc3 (ref. [77]) (https://docs.pymc.io/en/v3/index.html), ExoTEP[48–50], Batman[33], (http://lkreidberg.github.io/batman/docs/html/index.html), ExoTiC-ISM[54] (https://github.com/Exo-TiC/ExoTiC-ISM), ExoTiC-LD[55] (https://exotic-ld.readthedocs.io/en/latest/), Emcee[32] (https://emcee.readthedocs.io/en/stable/), Dynesty[68] (https://dynesty.readthedocs.io/en/stable/index.html) and chromatic (https://zkbt.github.io/chromatic/), which use the python libraries scipy[78], numpy[79], astropy[80,81] and matplotlib[82]. The atmospheric models used to fit the data can be found at PICASO[60–63] (https://natashabatalha.github.io/picaso/), Virga85 (https://natashabatalha.github.io/virga/), ScCHIMERA[58,59] (https://github.com/mrline/CHIMERA), ATMO[65,83] and PHOENIX[66,67].

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

**Acknowledgements** This work is based on observations made with the NASA/ESA/CSA JWST. The data were obtained from the Mikulski Archive for Space Telescopes at the Space Telescope Science Institute, which is operated by the Association of Universities for Research in Astronomy, Inc., under NASA contract no. NAS 5-03127 for JWST. These observations are associated with program no. JWST-ERS-01366. Support for program no. JWST-ERS-01366 was provided by NASA through a grant from the Space Telescope Science Institute, which is operated by the Association of Universities for Research in Astronomy, Inc., under NASA contract no. NAS 5-03127. This work benefited from the 2022 Exoplanet Summer Program in the Other Worlds Laboratory at the University of California, Santa Cruz, a program supported by the Heising-Simons Foundation. We thank E. Agol for constructive comments.

**Author contributions** All authors played a significant role in one or more of the following: development of the original proposal, management of the project, definition of the target list and observation plan, analysis of the data, theoretical modelling and preparation of this paper. Some specific contributions are listed as follows. N.E.B., J.L.B., Z.K.B-T., I.J.M.C., J.D., L.K., M.R.L., D.K.S., and K.B.S. provided overall program leadership and management. D.K.S., E.M.-R.K., H.R.W., I.J.M.C., J.L.B., K.B.S., L.K., M.L.-M., M.R.L., N.M.B., V.P. and Z.K.B.-T. made significant contributions to the design of the program. K.B.S. generated the observing plan with input from the team. B.B., E.M.R.K., H.R.W., I.J.M.C., J.L.B., L.K., M.L.M., M.R.L., N.M.B. and Z.K.B.-T. led or coled working groups and/or contributed to significant strategic planning efforts such as the design and implementation of the prelaunch Data Challenges. A.L.C., D.K.S., E.S., N.E., N.P.G., and V.P. generated simulated data for prelaunch testing of methods. Z.R., D.K.S., J.K., E.S., E.M.M. and A.L.C. reduced the data, modelled the light curves and produced the planetary spectrum. S.M., M.R.L., J.M.G. and J.L. generated theoretical model grids for comparison with the data. Z.R., D.K.S., C.P., E.M.M., E.S., J.K., M.L. and S.M. made significant contributions to the writing of this paper. Z.R., D.K.S., S.M., E.M.M. and M.R.L. generated figures for this paper. B.V.R., D.K.P., I.J.M.C., J.T., J.M.G., J.L.B., M.L.M., R.H., R.J.M., S.E.M. and T.D. contributed to the writing of this paper.

**Competing interests** The authors declare no competing interests.

**Additional information**
**Correspondence and requests for materials** should be addressed to Z. Rustamkulov.

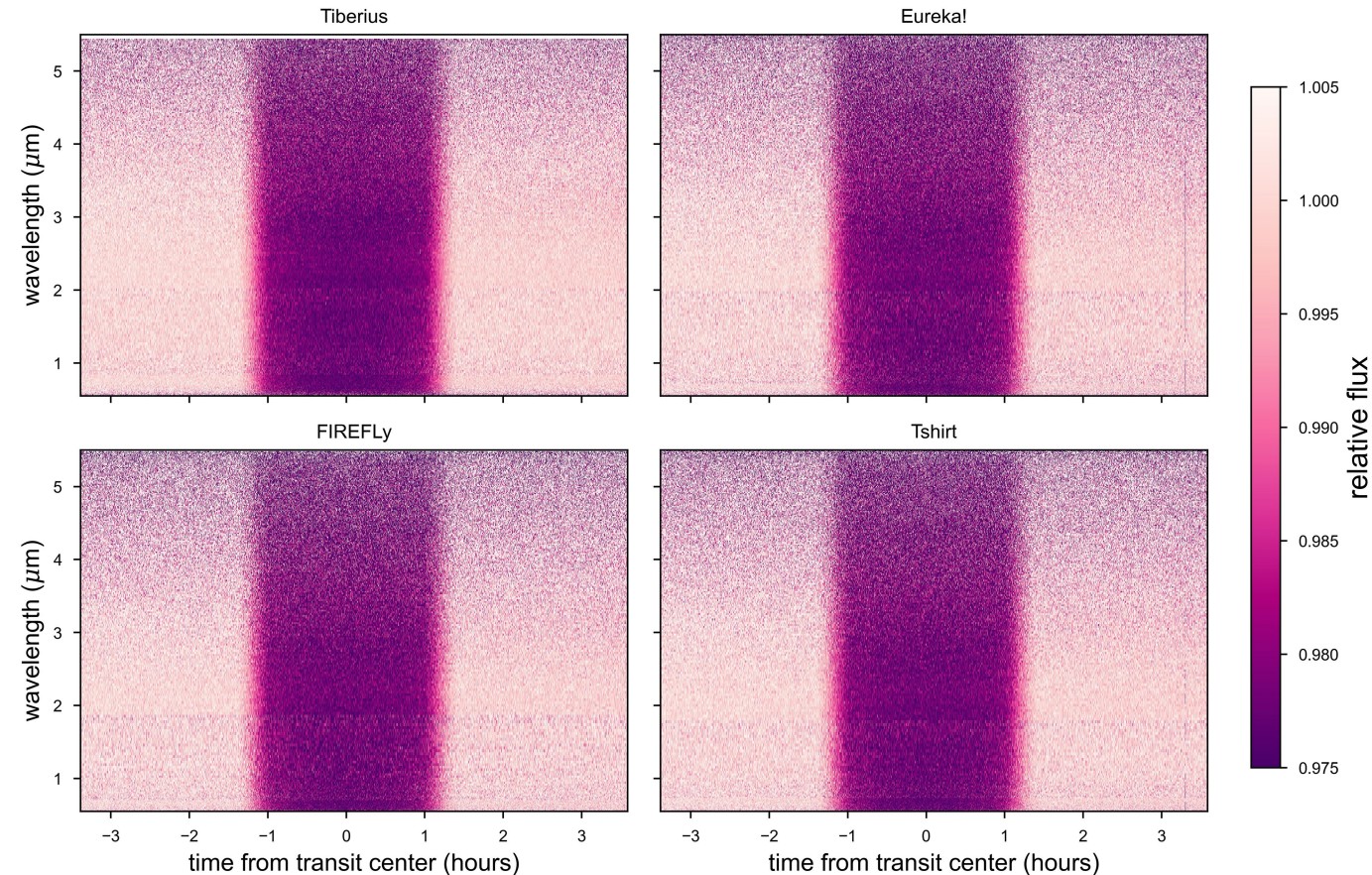

**Extended Data Fig. 1 | A comparison of the extracted 1D spectrophotometry across the four reductions.** Plotted is the spectrophotometry with time on the x-axis and wavelength on the y-axis, with color indicating the relative flux. The transit is visible as a dark band in the middle of the observation. All four reductions show nearly identical noise properties.

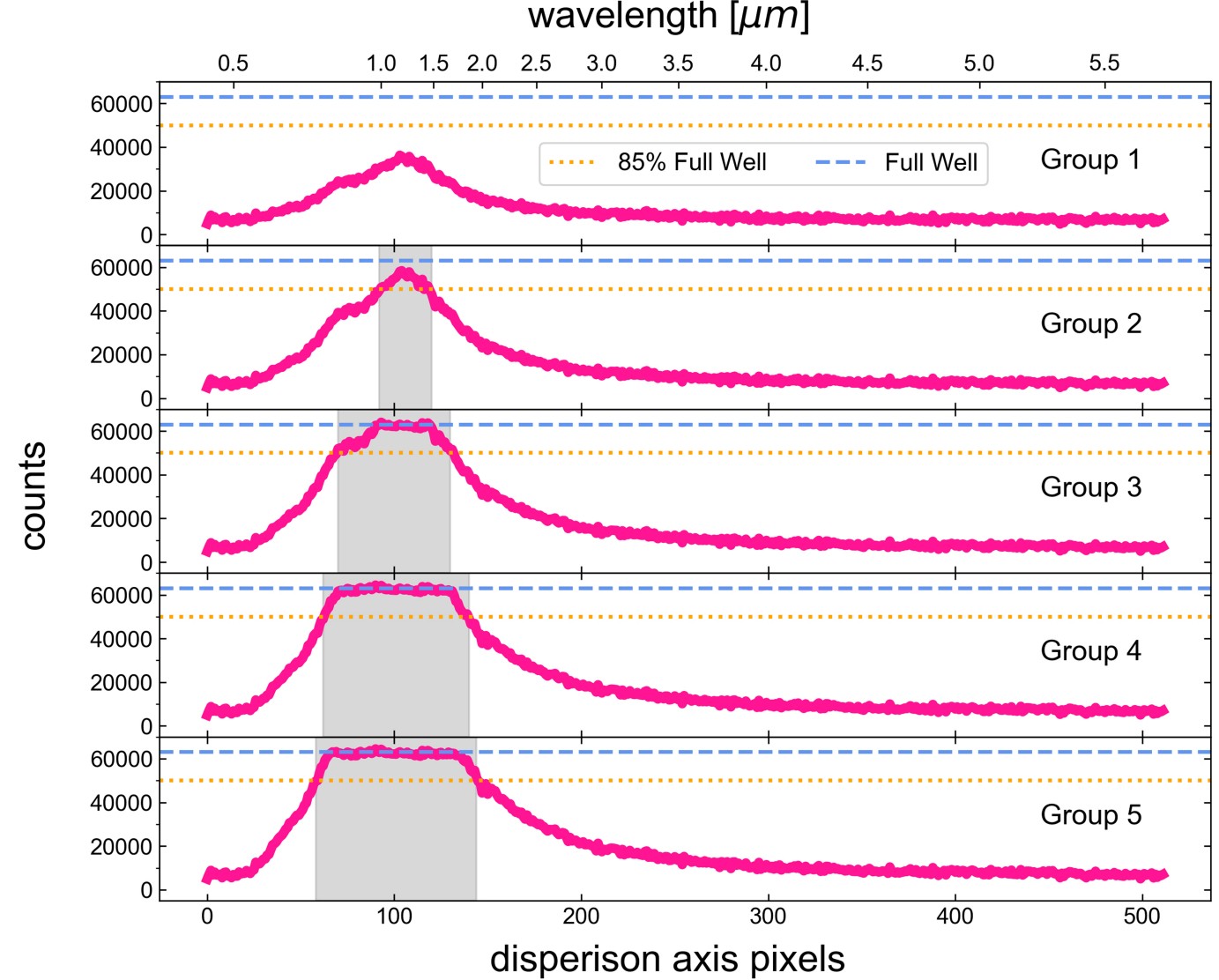

**Extended Data Fig. 2 | Demonstration of the impact of saturation.** Shown are the group-level median frames from the uncalibrated data products across the entire integration. The dashed blue line represents the empirically derived saturation level, with the orange dotted line representing 85% saturation, the level adopted in the Eureka! reduction. Grey shaded regions represent columns that reach 85% full well in a given group.

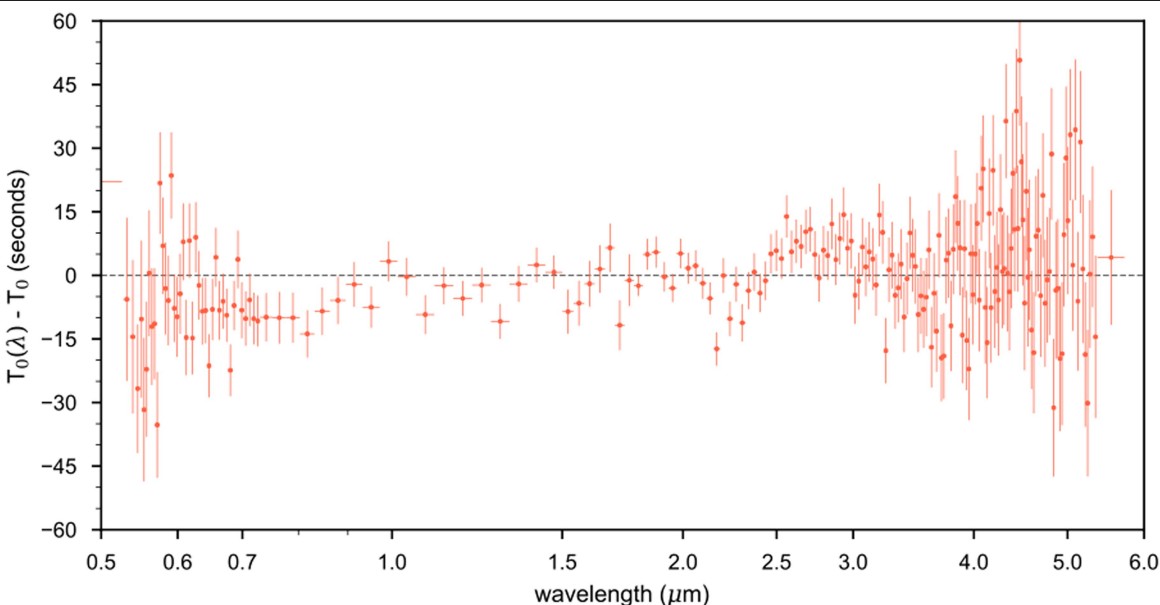

**Extended Data Fig. 3 | The wavelength-dependent central transit time in seconds.** Structure is apparent–the prominent water and carbon dioxide absorption features at 2.7 μm and 4.2 μm, respectively, appear to arrive ~20 s after the optical continuum. A slope is also apparent from the blue side to the red. The error bars are 1-$\sigma$ standard deviations.

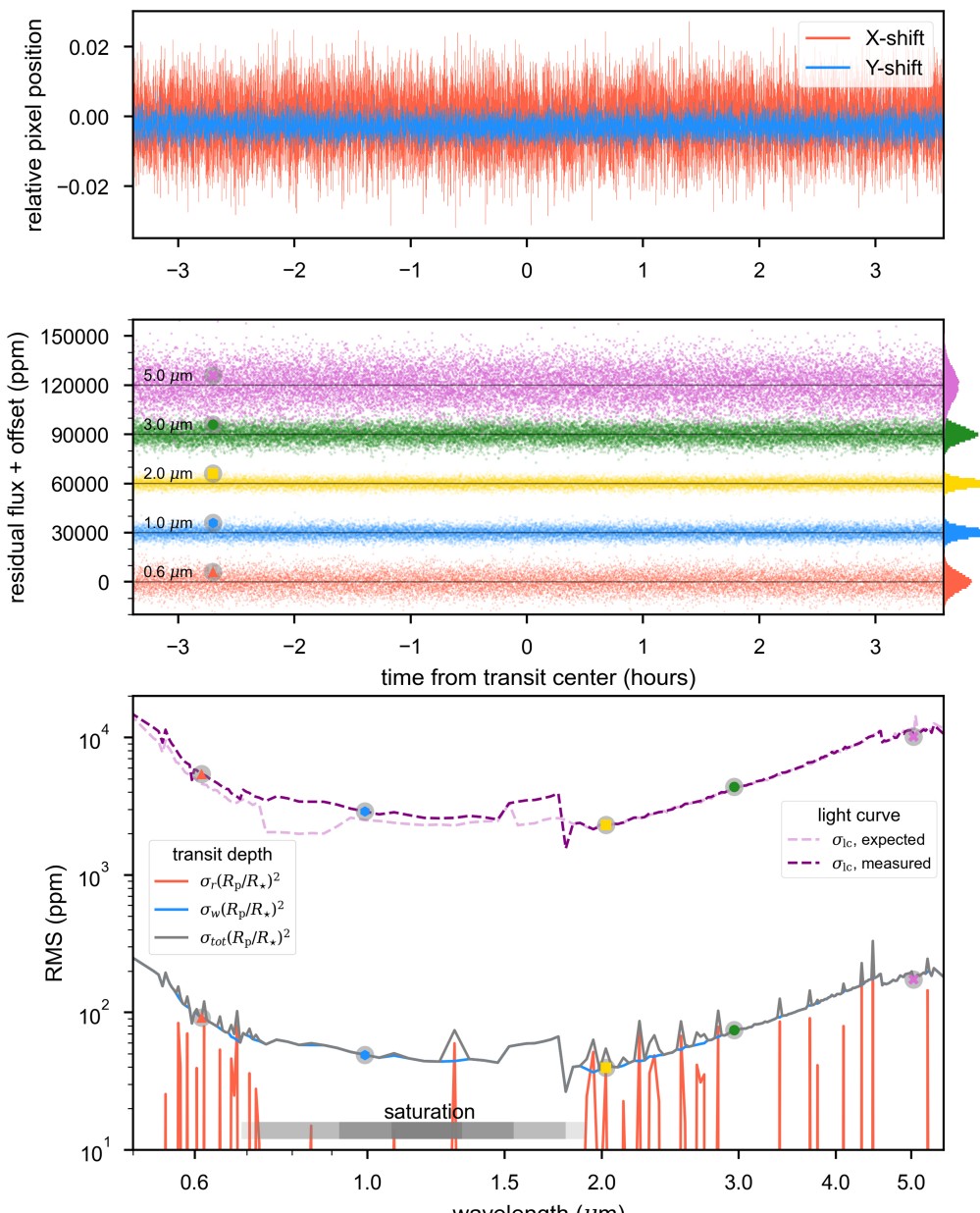

**Extended Data Fig. 4 | A summary of the positional shifts of the trace, the wavelength-dependent light curve scatter, and the transit depth noise.** (**Top**) The X- and Y-shift vectors as measured by 1D cross correlation with FIREFLy. (**Middle**) The residual spectrophotometric light curves are shown for four representative spectral channels spanning the PRISM wavelength range with no temporal binning. The residual scatter is approximately Gaussian for each, as indicated by the histogram on the right y-axis. We validate this by performing Anderson-Darling tests on the residuals of the spectral and white-light curves, and find that all of the Anderson-Darling test statistics lie below the respective critical values 1% significance level. Therefore, we find that there is not sufficient evidence that the residuals are not normally distributed. (**Bottom**) The top two purple curves show the expected and measured normalised light curve root mean square (RMS) residuals, with no temporal binning. Longward of 2 μm, the scatter in each light curve matches well with the expected noise as estimated by the jwst pipeline, which is dominated by photon noise. This agreement indicates the majority of the light curves reach near the photon limit. The transit depth uncertainties are also plotted below, including the white noise (blue, $\sigma_w$), red noise (red, $\sigma_{red}$), and total noise components (grey, $\sigma_{tot}$). Some wavelength bins have enhanced red noise, but the majority of the transmission spectrum is consistent with minimal red noise from residual systematic errors. The wavelengths affected by detector saturation are indicated by the grey shaded bar, with darker colors corresponding to quicker saturation. The colored dots are the measured RMS values from the light curves shown in the top panel.

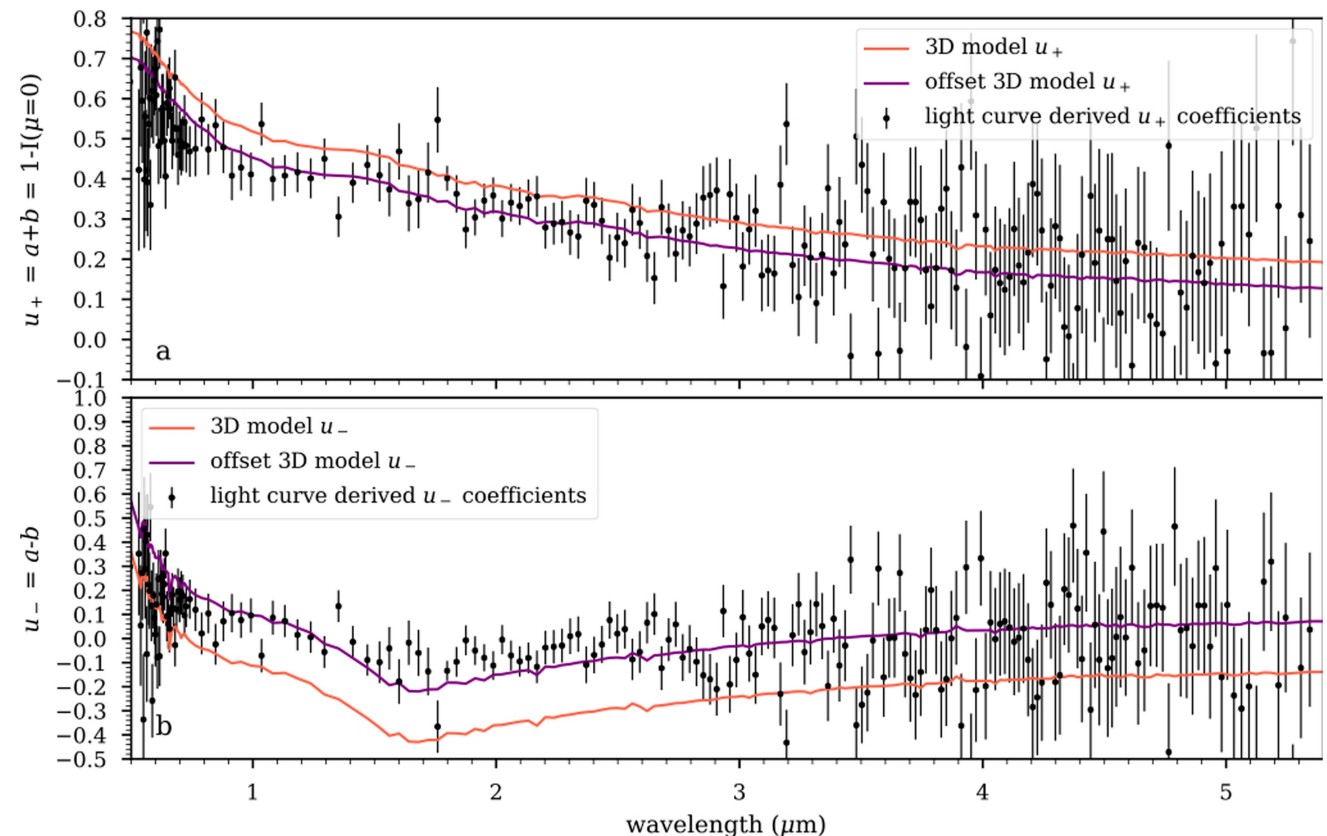

**Extended Data Fig. 5 | Empirically derived stellar limb darkening coefficients fit with a quadratic law. a**, the fit $u_+$ coefficients (black) along with the theoretically predicted values derived from a 3D stellar model (red). The theoretical $u_+$ values with a constant offset of $-0.065 \pm 0.022$ (purple) is also shown. The theoretical models predict the wavelength-to-wavelength shape of $u_+$ well. As $u_+$ is directly related to the intensity of the star at the stellar limb ref.[84], these findings suggest WASP-39A is 6% brighter at the limb than models predict. **b**, similar as **a**, but for the $u_-$ coefficient. As the shape of the derived coefficients differs from the model prediction, $u_-$ was left free to vary in the transmission spectral fits. The error bars are 1-$\sigma$ standard deviations.

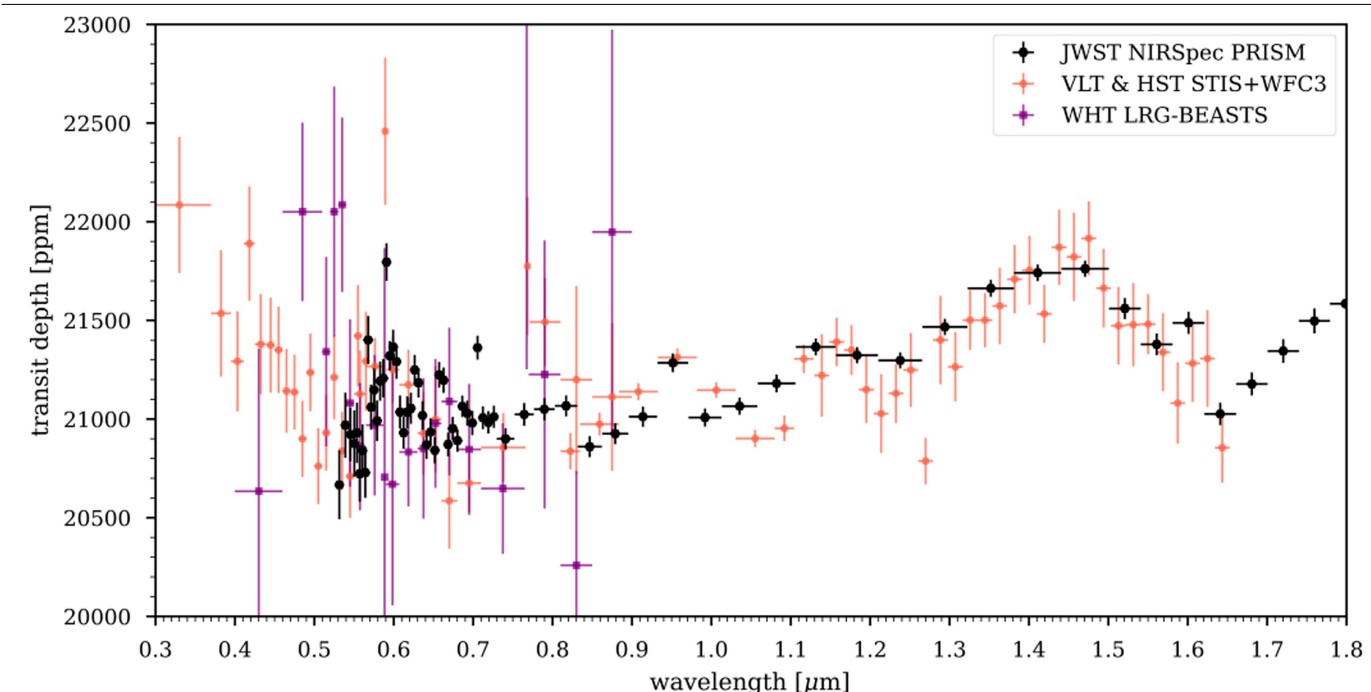

**Extended Data Fig. 6 | Comparison of the JWST NIRSpec PRISM data (black) to HST and VLT data from ref.** [6,15] **and WHT data from ref.** [16]**, respectively.** The JWST spectrum was derived with the limb darkening fixed to the same 3D stellar model as in[6] to aid comparisons. With fixed limb darkening, the JWST transmission spectrum has lower overall transit depths especially at optical wavelengths. The broadband spectrum from the two space telescopes compares well, including the amplitude of the 1.4 μm water feature first observed by HST/WFC3 and the Na feature near 0.6 μm observed by HST/STIS. The error bars are 1-$\sigma$ standard deviations.

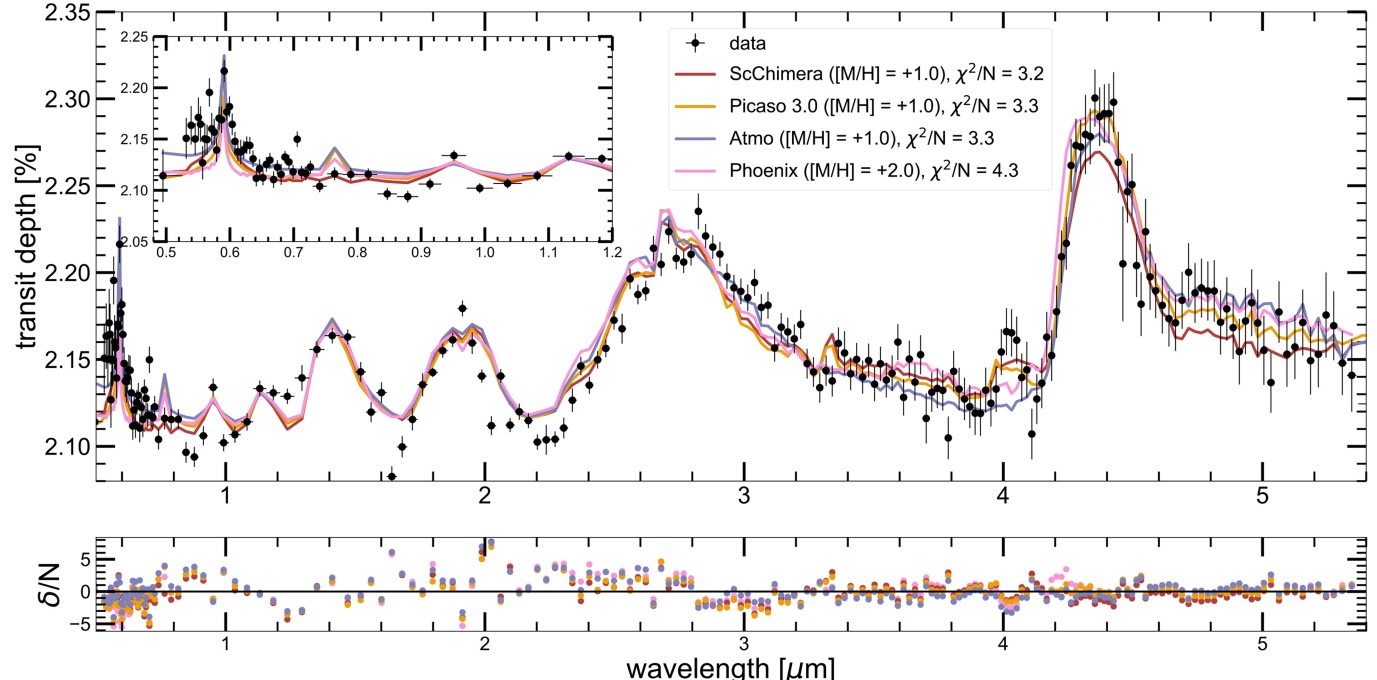

**Extended Data Fig. 7 | Best-fit models from ScCHIMERA, PICASO 3.0, ATMO, and Phoenix 1D RCTE model grids for WASP-39b.** The FIREFLy reduction is overlaid in the top panel. The top left inset panel shows the data and the models between 0.5–1.2 μm. All these models prefer super-solar atmospheric metallicities and cloudy atmospheres for WASP-39 b. The C/O ratio estimated by these models lies in the range 0.6– 0.7. Additional SO$_2$ was injected in the PICASO 3.0 and ScCHIMERA grids to estimate the abundance of SO$_2$ required to explain the 4.0 μm feature, in a Bayesian framework. The ATMO and PHOENIX models are shown without any additionally injected SO$_2$ to emphasize that RCTE models do not predict such an SO$_2$ feature and chemical disequilibrium effects are required to explain the observed feature. The bottom panel shows the residuals from each best-fit model divided by the noise in the transit depth as a function of wavelength. The error bars are 1-$\sigma$ standard deviations.

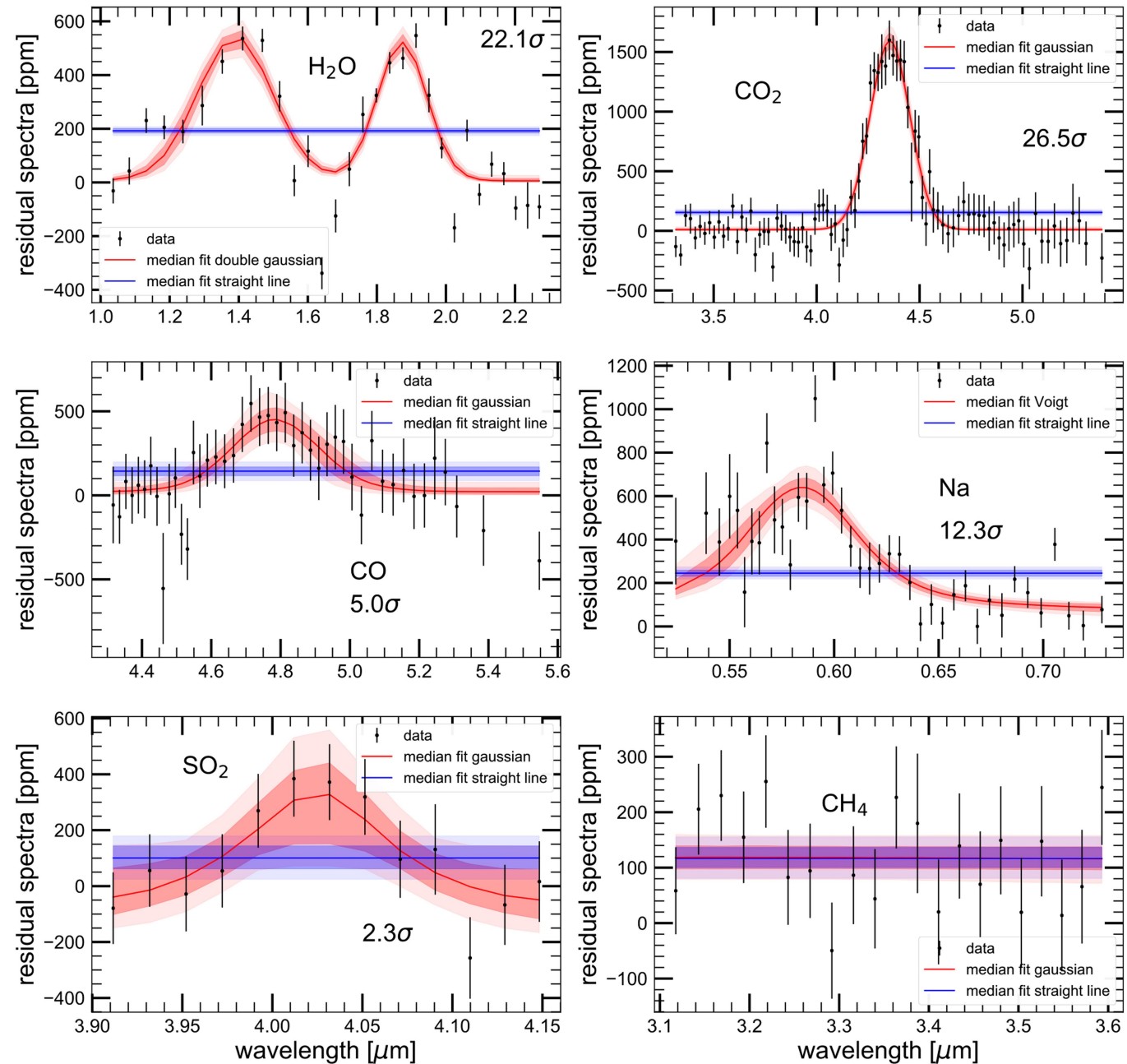

**Extended Data Fig. 8 | Each panel shows the residual spectrum of a particular gas.** This residual spectrum was obtained by removing one gas at a time from the best-fit model atmosphere and subtracting the recalculated model transmission spectrum without that gas from the data. This residual spectrum was then fitted with a Gaussian distribution (and a Voigt profile for Na) and a constant offset, in a Bayesian framework. The median fit (solid lines) along with the $1\sigma$ and $2\sigma$ confidence intervals are shown with shaded red and blue regions for the Gaussian fits and the constant offset fits, respectively. The Bayes factor between the two functional fits was used to determine the detection significance of each gas. Note that the wavelength range covered in each panel is different. The error bars are $1$-$\sigma$ standard deviations.

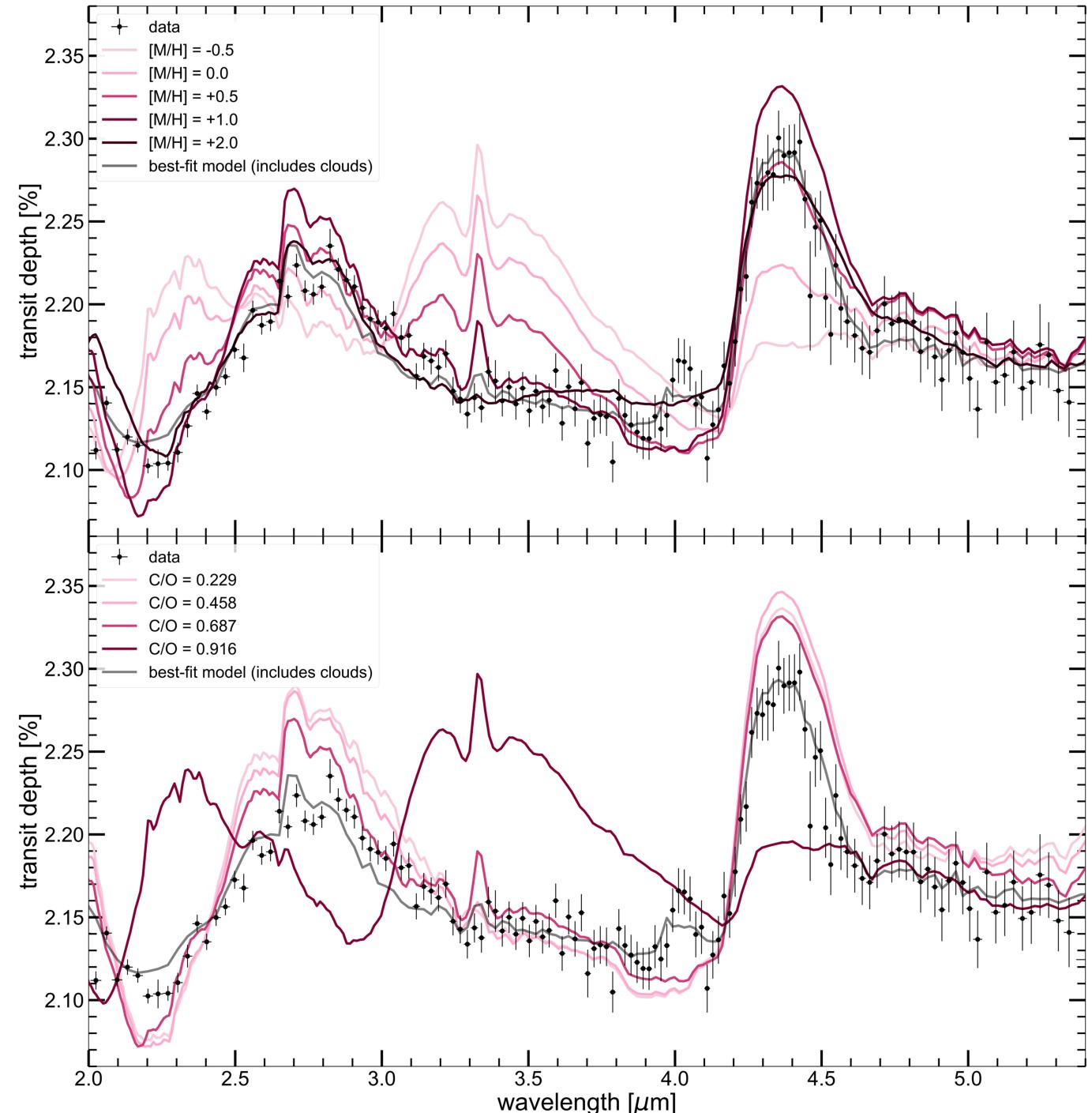

**Extended Data Fig. 9 | Models of varying metallicity (top) and C/O ratio (bottom) compared to the FIREFLy reduction.** A comparison of cloud-free PICASO 3.0 RCTE models across a span of metallicities with the best-fit C/O ratio (0.68) is shown in the top panel. Each line coloured from faded to deep pink represents models with different metallicities between sub-solar to super-solar values. The simultaneous lack of a prominent $CH_4$ feature at 2.3 and 3.3 μm and the presence of a strong $CO_2$ feature indicate that the observations disfavor a low-metallicity atmosphere. The bottom panel shows transmission spectrum models with different C/O ratios from sub-solar to super-solar values at 10×solar metallicity compared with the observed spectrum. The cloudy best-fit model obtained with the grid retrieval framework also has been shown in both the panels with the grey line. As before, the errorbars are 1σ standard deviations.

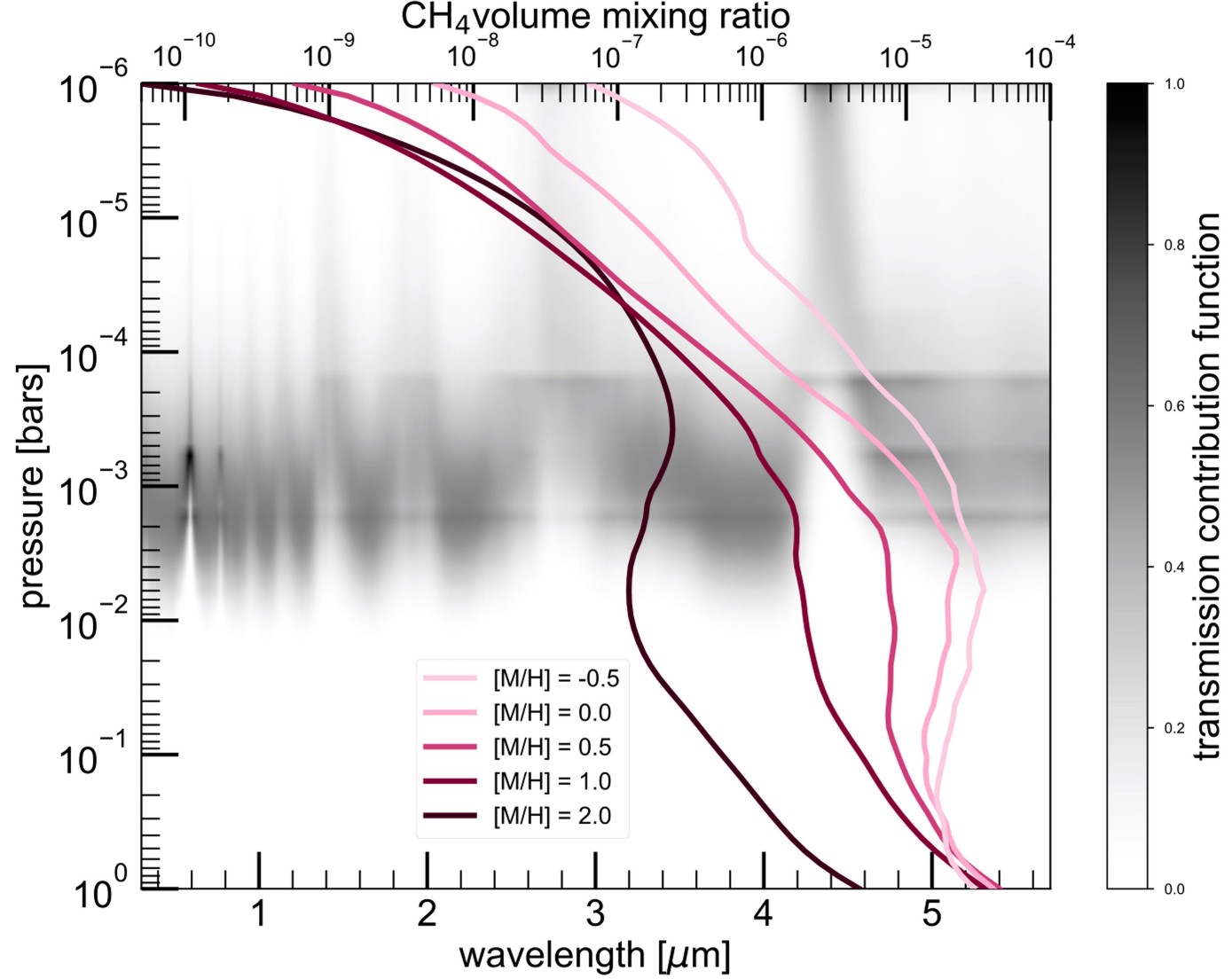

**Extended Data Fig. 10 | The wavelength-dependent contribution function.** The shaded regions highlight the parts of the atmosphere probed by the observed transmission data as a function of wavelength, as calculated from the best-fit model. This shows that the data mostly probe pressure ranges between 0.1 to 2 mbars. The $CO_2$ feature shows contribution at pressures approaching a microbar. The various shaded lines in pink show the volume mixing ratio of $CH_4$ (upper x-axis), from thermochemical equilibrium models, with different atmospheric metallicities at the best-fit C/O ratio of 0.68.

**Extended Data Table 1 | Best-fit orbital parameters as measured from the FIREFLy white light curve**

| Parameter | Value | Description |
|---|---|---|
| $T_0$ | $0.83566341 \pm 0.0000007$ | Mid-transit time [days] $(\mathrm{BJD_{TDB}} - 2459770)$ |
| $a/R_s$ | $11.405 \pm 0.011$ | Scaled semi-major axis |
| $b$ | $0.4458 \pm 0.0021$ | Transit impact parameter |
| $\varrho_s$ | $1.707 \pm 0.005$ | Stellar density [g cm$^{-3}$] (derived) |

The scaled semimajor axis and impact parameter are fixed when fitting for the transmission spectrum.

**Extended Data Table 2 | An overview of the analysis procedures used by the independent data reductions**

| Reduction step | FIREFLy | Tshirt | Eureka! | Tiberius |
|---|---|---|---|---|
| Background, $1/f$ subtraction | y | y | y | y |
| X-, Y-shift correction | y | n | y | y |
| X-, Y-shift detrending | y | n | y | y |
| Baseline detrending | y | y | y | y |
| Trace extraction optimization | y | y | y | n |
| Pre-transit baseline trim | y | y | n | y |
| **Mean spectrophotometric scatter (ppm)** | 676 | 725 | 815 | 709 |

The spectrophotometric scatter is estimated from the standard deviation of the pre-transit data between 0.62-5.42 μm with only a linear baseline trend removed.

**Extended Data Table 3 | Overview of the best-fit model parameters obtained from each grid**

| | PICASO 3.0 | | ScCHIMERA | | ATMO | PHOENIX |
|---|---|---|---|---|---|---|
| Parameter | w/ SR | w/o SR | w/ SR | w/o SR | w/ SR | w/ SR |
| [M/H] | +1.0 | +1.0 | +1.0 | +1.0 | +1.0 | +2.0 |
| C/O | 0.68 | 0.68 | 0.65 | 0.65 | 0.7 | 0.9 |
| $\varkappa_{cld}$ [cm$^2$/g] | $10^{-2.07}$ | $10^{-2.04}$ | $10^{-2.46}$ | $10^{-2.52}$ | $5 \times H_2$ | Opaque |
| $P_{cld}$ | – | – | – | – | 1-50 mbar | 1 mbar |
| Rayleigh Scattering | $H_2$ only | $H_2$ only | $H_2$ only | $H_2$ only | $10 \times$multigas | $H_2$ only |
| $\log_{10}(SO_2)$ | -5.2 | -5.1 | -5.7 | -5.7 | – | – |
| $\chi^2$/N | 3.3 | 3.2 | 3.2 | 2.9 | 3.3 | 4.3 |

PICASO 3.0 and ScCHIMERA grids follow the grid retrieval (GR) framework to obtain the best-fit models whereas ATMO and PHOENIX use the reduced $\chi^2$ minimization based grid search method (GS). To test the effect of the saturated region on the obtained best-fit parameters, the PICASO 3.0 and ScCHIMERA grid were used to also do a fit with the error bars in the saturated region (0.68 µm – 1.91 µm) inflated 1000 times. The best-fit parameters did not show any significant change due to this exercise but are still listed in the table under the w/o SR column. The best-fit parameters obtained by fitting the full spectrum are listed under the w/ SR column. Note that even though the w/o SR fits were obtained by inflating the errorbars in the saturated region, the reduced $\chi^2$ reported in the w/o SR column are computed without the points in the saturated region for direct comparison with the reduced $\chi^2$ obtained from fitting the full spectrum. Also, note that the ATMO models include cloud opacities with an adjustable multiple of the $H_2$ Rayleigh scattering opacity at 350 nm. Therefore the $5 \times H_2$ in this table for the ATMO grid corresponds to a gray cloud opacity which is $5 \times$ the $H_2$ Rayleigh scattering opacity at 350 nm between 1 to 50 mbar pressures.

**Extended Data Table 4 | Detection significances of individual opacity sources with our two techniques: Bayes factor analysis with gas removal, and Gaussian/Voigt fits to the residual absorption profiles[85–92]**

| Gas | Bayesian gas removal | | Residual fit | |
|---|---|---|---|---|
| | ln(B) | $\sigma$ | ln(B) | $\sigma$ |
| [75] $H_2O$ | 537.7 | 32.9 | 242.3 | 22.1 |
| [76] $CO_2$ | 374.3 | 27.5 | 348.9 | 26.5 |
| [77] $CO$ | 24.6 | 7.3 | 10.68 | 5.0 |
| [78] $H_2S$ | -44.9 | N/A | -0.4 | N/A |
| [79] $CH_4$ | -8.9 | N/A | -0.05 | N/A |
| [80] $SO_2$ | 2.2 | 2.7 | 1.6 | 2.3 |
| [81] Na | 173.9 | 18.8 | 73.2 | 12.3 |
| [82] K | 0.6 | 1.7 | -1.0 | N/A |
| Cloud | 209.8 | 20.6 | N/A | N/A |

Note, a negative ln(B) indicates that that specific opacity source is not preferred by the data.