## [Peer Review File · Nature]

Manuscript Title: Early Release Science of the exoplanet WASP-39b with JWST NIRSpec PRISM

Reviewer Comments & Author Rebuttals

Reviewer Reports on the Initial Version:

Referee #1 (Remarks to the Author):

The manuscript presents the JWST broadband transmission spectrum of WASP-39b from observation of one transit together with detection of multiple chemical species. The spectacular spectrum showcases the capabilities of JWST NIRSpec PRISM. Its presentation can serve as a benchmark for future exoplanet transmission spectroscopy programs. The analysis of the data leading to the (non-)detection of several chemical species is sound but could benefit from a more detailed discussion including relevant comparison to other planets and/or observations.

While the data presented are highly interesting and relevant, I am hesitating to recommend the manuscript for publication, mainly because ref. [22] points to a forthcoming article in nature by the same authors that shows the same spectrum but limited to 3-5.5 μ m. That paper also reports the presence of several chemical species. Therefore, I believe the two manuscripts are partially redundant, and I am not convinced that it would serve the reader to publish both as articles in nature.

The references in the paper could benefit from including other work in transmission spectroscopy, in particular from ground based observations. Figs. 1 and 2 look convincing but transport little scientific information rather than the data quality that is nicely visible in the upper panel of Fig.1 already. Fig.3 is confusing because it plots the same data several times but using different reduction methods. This plot could go to the Methods part. Fig.4 appropriately presents the data and the model contributions from different species.

The discussion of the results in scientific context is very short. The main part of the manuscript largely avoids mentioning specific results and concentrates on the overall usefulness of the JWST spectra, which I believe is out of question, in particular after the publication of ref.[22]. I am not sure what the authors mean by "the limited number of groups available to estimate the flux within a detector integration warrants caution..." (first half of last paragraph). The last three paragraphs of the main text should be checked for clarity and focus.

In the Methods part of the manuscript, the authors include discussions of all

relevant parts of the methodology. Here, the significance of the detections of molecular species is of particular importance and could provide a benchmark for the achievable precision in transmission spectroscopy. Also, a comparison to other methods or data sets would be useful as well as an estimate for the significances achievable in other planets, particularly very small ones.

Referee #2 (Remarks to the Author):

This is an important paper, and is highly appropriate for Nature. The wavelength span that was observed is impressive. I have no major doubts concerning the results, but I have five significant issues that should be addressed, and some minor comments:

Issue 1: The paper says very little about the spectrum of the host star, except to note that it is "relatively inactive". I suggest quoting some quantitative index of activity. More important is the expectation that the star has an absorption line spectrum of its own, and some thought should be given to how that affects the results. Specifically, the star will have carbon monoxide absorption lines (because the Sun does, and this star is cooler than the Sun). Those stellar lines could be quite strong in the fundamental band near 4.7 microns, and they will overlap the CO lines in the planet (because the orbit of the planet is close to circular). That can lead to a bias in the transit spectrum (explained in ApJ, 841, L3). If that's an important effect in this case, it needs to be accounted for in the analysis.

Issue 2: I am impressed with the work that the authors have done to correct for the detector saturation. In addition to that, I suggest doing the model fits by simply omitting (zero-weighting) the region of saturation. Since that region contains only weak water absorption, the parameters of the best-fit spectrum probably won't change very much, but demonstrating that would increase confidence in the results.

Issue 3: The paper uses multiple data analysis pipelines to derive transit spectra. Those pipelines have different procedures for issues such as background subtraction, baseline slopes, limb darkening fitting, treatment of X- and Y-coordinate shifts, etc. It is encouraging that the derived spectrum does not depend strongly on the details, but the differences are confusing to the reader. It would be very helpful to include a summary table, indicating at-a-glance how the different pipelines treat the various issues. It would also be helpful to include some numbers for what fraction of the photoelectron-limited signal-to-noise-ratio is achieved by each pipeline.

Issue 4: The conceptual basis for fitting the forward model grids needs a clearer explanation. Since you're invoking Bayes factors for the various gases, a reader may assume that the forward grids were fit using a Bayesian formalism. However, it seems like you're just minimizing chi-squared when fitting the full models. I think that's fine, because (as you point out) a rigorous Bayesian fit would require adopting priors based on other physical processes such as photochemistry that are best left to future work. But please be clear on the conceptual basis for fitting the model grids, i.e. you are minimizing chi-squared, or something more sophisticated. Also, it would be very helpful to have a

summary Table (beyond Table 2) giving the best-fit metallicity and C/O from each grid, and that would also be a good place to list the metallicity results when the saturated region is omitted.

Issue 5: I'm worried about the procedure of fitting Gaussian profiles to each molecular species (Figure 12). Molecules often have band heads, and rotational temperatures that produce non-Gaussian band shapes, albeit those shapes are affected by instrumental resolution. You have real band shapes that are plotted on Figure 4, and those seem more appropriate than Gaussian fits. Especially for the SO₂ case, I am worried that the real band shape might disagree with the data. The SO₂ opacity on Figure 4 appears to have a band head (sharp edge) at the short wavelength side, and that might disagree with the data if you were to fit it on Figure 12. If so, we need to know that.

Minor comments:

The paper should clearly describe the source of the wavelength calibrations, i.e. how wavelengths were assigned to columns. I don't find a clear description of that process, only vague comments.

When describing signal levels, the paper mentions "counts". That isn't a useful physical term; please quote signal levels in electrons. Also, the paper does not clearly describe what fraction of the theoretical (photoelectron-limited) signal-to noise ratio is obtained. Please be very clear about that; it would be helpful to include it in a summary table.

There are some technical issues that need better explanation: the 1/f noise - what it is and (if known) what causes it needs an explanation. Also what are bias frames and what are they used for, and similarly what are reference pixels? (The paper does a good job of explaining the sample-up-the-ramp non-destructive read process, but the 1/f noise, bias frames, and reference pixels need similar clarity.)

In the Fig 1 caption - how large is the linear trend that was removed? (Other investigators may want to compare to data in other programs.)

When fitting the white-light curve, non-Gaussian degeneracies in the posterior distributions are mentioned. Posteriors are mentioned, but is this really a Bayesian fit? MCMC is just a sampler, it doesn't imply that the fitting process is Bayesian. If the fit is Bayesian, what are the priors? (There can be no posteriors without priors). I don't think a Bayesian fit is needed, but a clear description of the principles of the fit is definitely needed.

Figure 5 is a good idea, but the current version isn't useful, because differences are too small to see. Consider subtracting a mean frame and showing the differences on an expanded scale.

The phrase "used them to shift stabilize the images with flux-conserving interpolation." needs a better explanation. (Stabilizing an image sounds like something that is done by hardware, not by data analysis.)

"We chose a 4-pixel-wide aperture to encompass the peak of the stellar flux without unnecessary

background." If background is significant, then an optimally-weighted extraction should be done as per the standard recipe (PASP, 98, 609). If background is negligible compared to this bright star, say so explicitly (and use a wider window to gather maximum signal). Background subtraction is an important technical issue and differences among pipelines should be highlighted in a summary table.

"We calculate uncertainties in the stellar spectra from the photon noise (before the background subtraction step)" That statement raises questions about precisely how that was done, and was it a correct procedure. Please explain more clearly, using an equation if needed.

Regarding Figure 9 and the statement "This offset suggests the limb of WASP-39A is brighter than the stellar models predict." Please re-visit this in more depth. Typically, quadratic functions don't fit model atmosphere intensities very well near the limb. So if they don't fit models very well, then of course they won't fit the real star very well either. I'm suggesting that the problem could be with the quadratic approximation, not the model atmospheres. Showing a quadratic fit to the model's center-to-limb intensities could help to clarify this issue.

In the caption of Figure 8, it would be helpful to subsequent investigators to be more specific concerning the properties of the noise. Instead of saying "The residual scatter is approximately Gaussian", please be quantitative and quote the results of Anderson-Darling tests, or similar normality statistics.

When describing the tshirt spectral extractions, the statement "No corrections were made to the centroid or wavelength solution due to the exceptional pointing stability of the observatory" seems to contradict a previous statement that "We measure x- and y-jitter systematics at the ~100 ppm level." I realize that different pipelines will make different approximations, but either the X- and Y-shifts are significant for extracting the transit spectrum, or they're not. Please be clear about whether position-related corrections are needed. It would be helpful to plot the X and Y shifts as a function of time - you already have 25 time series plots on Fig 2, so I'm confident you can find room to plot X- and Y- time series somewhere. And please describe (in a summary table) how different pipelines treat this issue.

In the tshirt section, "We also assume an exponential ramp in time to the data", be careful using the word "ramp". The Spitzer investigators used it to mean a temporal baseline spanning much of the transit. JWST detectors use a sample-up-the-ramp procedure, and that's a different usage of ramp. I suggest saying "temporal baseline" when that's what you mean. (The term "ramp" wasn't confusing in the Spitzer era, but it could be now.)

In the Tiberius section, you describe "replacing pixels containing a cosmic ray with a linear interpolation between neighboring pixels." Strictly speaking, replacing pixels is an incorrect procedure (you're manufacturing data that you don't have, based on unverified assumptions). The strictly correct procedure is to zero-weight the bad pixels. In actual practice, it probably doesn't matter, but it would be helpful to describe some tests to ensure that your transit spectrum is not sensitive to this procedure.

As regards your best-fit grey cloud opacity, it is very interesting and would be worth noting that the

value you derive is close to the grey opacity inferred by early HST spatial scan spectroscopy of water vapor in HD209458b (0.012 cm²/g, see ApJ, 774, 95).

solar should not be capitalized. The Sun is capitalized, but not solar.

Author Rebuttals to Initial Comments:

Author's response:

We thank the referees for their insightful and timely review of the paper, and addressing the comments have improved the manuscript. The updated manuscript reflects the changes suggested by the referees, and we have made several additional changes as well. As requested by the editor, we have shortened the main text to better fit in the guidelines of Nature. We have also re-computed several of the model grids, as we noticed the strong CO₂ feature was artificially muted at the highest metallicities by the previous top of the atmosphere pressure level set. The new models do not change the overall best fit, so the conclusions are the same, but the methods Figure 13 was updated. We have also updated the title to better match the other JWST ERS paper efforts under submission to Nature.

Referees' comments:

Referee #1 (Remarks to the Author):

The manuscript presents the JWST broadband transmission spectrum of WASP-39b from observation of one transit together with detection of multiple chemical species. The spectacular spectrum showcases the capabilities of JWST NIRSpec PRISM. Its presentation can serve as a benchmark for future exoplanet transmission spectroscopy programs. The analysis of the data leading to the (non-)detection of several chemical species is sound but could benefit from a more detailed discussion including relevant comparison to other planets and/or observations.

While the data presented are highly interesting and relevant, I am hesitating to recommend the manuscript for publication, mainly because ref. [22] points to a forthcoming article in nature by the same authors that shows the same spectrum but limited to 3-5.5 μ m. That paper also reports the presence of several chemical species. Therefore, I believe the two manuscripts are partially redundant, and I am not convinced that it would serve the reader to publish both as articles in nature.

Response:

The WASP-39b JWST NIRSpec prism data has a number of high-profile science results, capabilities, and rich diversity of exoplanetary spectral features which haven't been observed together before. Thus, we feel there is room for multiple publications beyond letters reporting detections of a newly discovered species. The initial prism paper covering CO₂ was intended to be published as quickly as possible after the data were taken, as it had never been detected before in an exoplanet atmosphere and highlighted JWST's capabilities to detect a scientifically important molecule. The rest of the rich prism spectra is presented here, and contains further novel results. Exoplanetary carbon chemistry has not been fully explored to date, but with the prism spectrum, one is able to simultaneously study CO, CO₂, CH₄ and H₂O along with optical clouds and alkali metals. We feel presenting the results of entire spectra, the detection significances for the full suite of molecules (rather than only CO₂ as in ref 22), and the resulting chemical constraints help set the manuscripts apart. In addition, this manuscript presents novel evidence

of photochemistry in the detection of SO₂, as well as molecular signatures of limb asymmetries in H₂O and CO₂ which have not been previously detected. We have updated the discussion of the manuscript (as suggested in a comment addressed below) to better highlight these new scientific findings.

More generally, for the Transiting ERS program on WASP-39b, we have transits with multiple instruments and configurations including NIRSpec prism in this work and NIRSpec G395H which is being published separately. The overall plan is to compare modes, performance, and resulting science that is best suited to the modes and instruments, but to get the works out quickly to be informative to the community both initial works had to be done in parallel led by a different set of authors. The G395H mode will also have an overlapping wavelength range with prism. Between G395H and prism, the two main differences are the wavelength coverage and resolution, with the prism having very wide coverage from the optical to 5.5 microns, but a much lower resolution than the G395H. While the two papers do indeed detect similar overlapping features, there are unique aspects to both, and we have made deliberate efforts for the whole suite of papers to highlight scientific results which best suit that particular mode/instrument. A subsequent paper is in progress to synthesize these datasets and report on final combined constraints such as the planet's metallicity.

The references in the paper could benefit from including other work in transmission spectroscopy, in particular from ground based observations. Figs. 1 and 2 look convincing but transport little scientific information rather than the data quality that is nicely visible in the upper panel of Fig.1 already. Fig.3 is confusing because it plots the same data several times but using different reduction methods. This plot could go to the Methods part. Fig.4 appropriately presents the data and the model contributions from different species.

Response:

We have included Figures 1 and 2 as we feel the overall data quality from a light curve level is important to get across and is standard in the field. For example, variations of these plots are shown in Kreidberg et al. (2014, Nature, 505, 69), de Wit et al. (2018, Nature Astro, 2, 2014), Tsiaras et al. (2019, Nature Astro, 3, 1086), Spake et al. (2018, Nature, 557, 68). As JWST is a new facility, we feel it is important in these initial papers to fully display the data quality resulting from this facility. We also think that including different independent data analysis is also important, as the overall agreement is very high and helps the reader know that the planetary features are independent of the methods used. This has not historically always been the case with exoplanet atmosphere detections.

The transmission spectra is compared to two ground-based results, the VLT data from Nikolov et al. (2016) which was combined with the HST data and published in Wakeford et al. (2018) and the WHT data from Kirk et al. (2019). We have clarified the caption and legend of Figure 10 in the methods to make this more clear, and added references into the main text as well which now reads:

"The transmission spectrum also agrees well with previous measurements from ground-based telescopes[15, 16] as well as HST and Spitzer[6] within error (see Fig. 3), indicating that we are able to reliably recover a spectrum at these levels of saturation."

The discussion of the results in scientific context is very short. The main part of the manuscript largely avoids mentioning specific results and concentrates on the overall usefulness of the JWST spectra, which I believe is out of question, in particular after the publication of ref.[22]. I am not sure what the authors mean by "the limited number of groups available to estimate the flux within a detector integration warrants caution..." (first half of last paragraph). The last three paragraphs of the main text should be checked for clarity and focus.

Response:

We agree with the referee on the scientific content in the main paper, which was largely buried in the methods section. From the main text, we have removed much of the text describing the usefulness, which we note the editor also requested and which also helped with the word limits. We shortened the text regarding the discussion with saturation, moving it to earlier in the article where the data are discussed. Finally, we added a new discussion about the scientific results in the final paragraph of the main text.

In the Methods part of the manuscript, the authors include discussions of all relevant parts of the methodology. Here, the significance of the detections of molecular species is of particular importance and could provide a benchmark for the achievable precision in transmission spectroscopy. Also, a comparison to other methods or data sets would be useful as well as an estimate for the significances achievable in other planets, particularly very small ones.

Response:

For other planets, it is relatively difficult to make broad statements about what is achievable for small planets as it'll very much depend on the observability of the target and model atmospheres assumed. However, we can remark on transmission spectra signal-to-noise estimations, which the field largely uses PandExo to calculate for a given planet and atmospheric model (Batalha et al. 2017, PASP, 129, 976). Figure 8 in the methods indicates that both the measured light curves, and final transits depths are very close to the expected limits (photon noise + readout noise). Therefore, from the prism data here, tools such as PandExo should accurately predict the signal-to-noise levels and therefore what is achievable in other planets. We added the following sentence to the methods:

"As the noise levels are very close to the limit with what is expected including only photon and read noise sources, tools such as PandExo (Batalha,2017,PASP, 129, 4501B) should accurately predict what is achievable in other planets."

Referee #2 (Remarks to the Author):

This is an important paper, and is highly appropriate for Nature. The wavelength span that was observed is impressive. I have no major doubts concerning the results, but I have five significant issues that should be addressed, and some minor comments:

Issue 1: The paper says very little about the spectrum of the host star, except to note that it is "relatively inactive". I suggest quoting some quantitative index of activity. More important is the expectation that the star has an absorption line spectrum of its own, and some thought should be given to how that affects the results. Specifically, the star will have carbon monoxide absorption lines (because the Sun does, and this star is cooler than the Sun). Those stellar lines could be quite strong in the fundamental band near 4.7 microns, and they will overlap the CO lines in the planet (because the orbit of the planet is close to circular). That can lead to a bias in the transit spectrum (explained in ApJ, 841, L3). If that's an important effect in this case, it needs to be accounted for in the analysis.

Response:

For the stellar activity, the ERS team monitored the star with the NGTS network and analyzed TESS data as well. These analyses are given in an accompanying WASP-39b NIRCcam paper, which we have cited (Ahler et al. 2022, Nature, submitted). The NIRCcam paper reports low photometric activity levels from NGTS monitoring showing a 0.06% modulation, and the TESS data lack any star-spot crossings. They concluded, "WASP-39 is a quiet star and that the JWST ERS transit observations are unlikely to be adversely affected by stellar variability." We include these details in the Methods in a new short sub-section, citing the Ahler paper.

We thank the referee for pointing out the Resolution Linked Bias (RLB) effect. We estimated the effect with high resolution Phoenix models, both for the star and best-fit planet model. Following the procedure in ApJ, 841, L3, we find the effect is small but potentially significant for CO, peaking at about 30 to 40 ppm (see Response Fig 1 below). With about 25 data points affected around the 4.7 micron CO feature, each with ~150 ppm error bars, the RLB effect is significant in our data at about the 1-sigma level. We have added as subsection to the Methods to discuss this effect, citing the aforementioned ApJ paper and providing the estimate of 30-40 ppm for the 4.7 micron CO bandhead.

Response Fig. 1. Resolution Linked Bias effect calculated for WASP-39b. Shown is the difference calculating the transmission spectra using equation 4 and 5 in ApJ, 841, L3. In the CO bandhead around 4.7 microns, the RLB effect is on the order of 40 to 50 ppm on the transit depths.

Issue 2: I am impressed with the work that the authors have done to correct for the detector saturation. In addition to that, I suggest doing the model fits by simply omitting (zero-weighting) the region of saturation. Since that region contains only weak water absorption, the parameters of the best-fit spectrum probably won't change very much, but demonstrating that would increase confidence in the results.

Response:

We inflated the errorbars in the saturated region 1000 times to zero-weight them and test whether our inferred metallicity and other parameters depend on the saturated region. We performed this test with both Picaso and ScCHIMERA in the grid-retrieval framework. We find that the best-fit parameters show no change as also expected by the referee. We have added a description of this test to the methods Forward Model Grids section, “*Table 3 provides a summary... when the saturated region error bars were inflated by a factor of 1000.*”

Issue 3: The paper uses multiple data analysis pipelines to derive transit spectra. Those pipelines have different procedures for issues such as background subtraction, baseline slopes, limb darkening fitting, treatment of X- and Y-coordinate shifts, etc. It is encouraging that the derived spectrum does not depend strongly on the details, but the differences are confusing to the reader. It would be very helpful to include a summary table, indicating at-a-glance how the different pipelines treat the various issues. It would also be helpful to include some numbers for what fraction of the photoelectron-limited signal-to-noise-ratio is achieved by each pipeline.

Response:

We thank the referee for the suggestion and now have included the suggested Table 2 in the Methods, appended below.

Reduction step	FIREFLy	Tshirt	Eureka!	Tiberius
Background, $1/f$ subtraction	y	y	y	y
X-, Y-shift correction	y	n	y	y
X-, Y-shift detrending	y	n	y	y
Baseline detrending	y	y	y	y
Trace extraction optimization	y	y	y	n
Pre-transit baseline trim	y	y	n	y
Mean spectrophotometric scatter (ppm)	676	725	815	709

Table 2 An overview of the analysis procedures used by the independent data reductions. The spectrophotometric scatter is estimated from the standard deviation of the pre-transit data between 0.62-5.42 μm with only a linear baseline trend removed.

Issue 4: The conceptual basis for fitting the forward model grids needs a clearer explanation. Since you're invoking Bayes factors for the various gases, a reader may assume that the forward grids were fit using a Bayesian formalism. However, it seems like you're just minimizing chi-squared when fitting the full models. I think that's fine, because (as you point out) a rigorous Bayesian fit would require adopting priors based on other physical processes such as photochemistry that are best left to

future work. But please be clear on the conceptual basis for fitting the model grids, i.e. you are minimizing chi-squared, or something more sophisticated. Also, it would be very helpful to have a summary Table (beyond Table 2) giving the best-fit metallicity and C/O from each grid, and that would also be a good place to list the metallicity results when the saturated region is omitted.

Response:

The reported fits are from a single best fit (from any given model grid) derived from multiple different ways. For the Bayes factor analysis used to quantify the detection significance, indeed a fully Bayesian approach via Nested Sampling was used to determine the Bayesian evidence for each nested model (gases removed). These points/differences are already clarified in the methods section under “Forward Model Grids”:

“While the ATMO and the PHOENIX grids were used to fit the data with a reduced χ^2 based grid search method, the `PICASO 3.0` and `ScCHIMERA` grids were used in a grid retrieval framework using a nested sampler^{{cite{speagle2020,Buchner2014}}}. Within each nested sample likelihood calculation, the transmission spectra are generated on-the-fly by post-processing the pre-computed 1D RCTE model atmospheres”

and in the “Detection Significances of Various Gases” section:

“We quantify the detection significance of each species through a Bayes factor analysis^{cite[e.g.,][Benneke2013]}. To do so within the `ScCHIMERA` grid retrieval framework, we remove each gas during the transmission spectrum computation step (the 1D RCTE atmosphere models remain unchanged) one at a time and re-run the nested sampler. We compare the Bayesian evidence of each removed-gas run to that of the grid retrieval with all of the gases. There is no change in the number of parameters with the exception of the cloud and SO_2 mixing ratio parameters.”

To help clarify these differences, we have added Table 3 as the referee suggests.

Parameter	PICASO 3.0		ScCHIMERA		ATMO	PHOENIX
	w/ SR	w/o SR	w/ SR	w/o SR	w/ SR	w/ SR
[M/H]	+1.0	+1.0	+1.0	+1.0	+1.0	+2.0
C/O	0.68	0.68	0.65	0.65	0.7	0.9
$\kappa_{cl d}$ [cm ² /g]	$10^{-2.07}$	$10^{-2.04}$	$10^{-2.46}$	$10^{-2.52}$	–	–
$\log_{10}(SO_2)$	-5.2	-5.1	-5.7	-5.7	–	–
χ^2/N	3.3		3.2		3.3	4.3
Method	GR	GR	GR	GR	GS	GS

Table 3 Best-fit parameters from each grid. GR = Grid-retrieval, GS = Grid Search.

Issue 5: I'm worried about the procedure of fitting Gaussian profiles to each molecular species (Figure 12). Molecules often have band heads, and rotational temperatures that produce non-Gaussian band shapes, albeit those shapes are affected by instrumental resolution. You have real band shapes that are plotted on Figure 4, and those seem more appropriate than Gaussian fits. Especially for the SO₂ case, I am worried that the real band shape might disagree with the data. The SO₂ opacity on Figure 4 appears to have a band head (sharp edge) at the short wavelength side, and that might disagree with the data if you were to fit it on Figure 12. If so, we need to know that.

Response:

Certainly we agree with the referee that a Gaussian or even Voigt profile may not be an adequate match to these features. We use the Voigt/Gaussian fitting method simply to determine if there is a significant signal present without having to identify the molecular absorber. In general, this fitting procedure provides reasonable fits to these residual features (see Methods Figure 12). To remedy the shortcomings of this method, we also provide an alternative detection quantification method based upon the Bayes factor method by removing the identified gas in question. This fully encompasses the spectral shape of the molecule in question across the full wavelength range. This procedure is described in the methods section: "Detection Significance of Various Gases". Table 4 demonstrates that the detection significances from both methods are qualitatively similar. For example, a strong detection is a strong detection and a weak detection is a weak detection regardless of the method.

Minor comments:

The paper should clearly describe the source of the wavelength calibrations, i.e. how wavelengths were assigned to columns. I don't find a clear description of that process, only vague comments.

Response:

We have added text to the methods to clarify the source of the wavelength calibration. Currently, the wavelength solution is derived from pre-flight ground-based calibration which covers a predefined subset of the sub-array, with the actual wavelength range of the on-sky prism data wider than the official STScI pipeline calibration, especially at the blue end. We extrapolated the wavelength solution to cover these bluer pixels, and performed tests to determine if there were any offset in the overall solution, which we did not find. The following text was added:

"To obtain our final wavelength calibration, we extrapolated the STScI-provided in-flight instrumental wavelength calibration data product across the detector edge pixels which did not have an assigned wavelength. The calibration was derived using the ground-based wavelength solution. We performed tests to search for zero-point offsets in the calibration versus the planetary and stellar spectra, and did not find any at the level of half a pixel width or greater."

When describing signal levels, the paper mentions "counts". That isn't a useful physical term; please quote signal levels in electrons. Also, the paper does not clearly describe what fraction of the theoretical (photoelectron-limited) signal-to noise ratio is obtained. Please be very clear about that; it would be helpful to include it in a summary table.

Response:

We note that after fitting up the ramp in the JWST STScI pipeline, the units of the spectra are in DN/sec, thus many of the reduction analysis worked in these units which were referred to as counts or counts/sec. The gainscale step does convert these values to electrons, though that is not propagated through to the fits file headers. We have added the following sentence in the methods to clarify the issue:

“We note that since the GAINSCALE step of the JWST pipeline applies a gain correction to the raw count rate files, the counts and count rates quoted herein are in units of electrons and electrons per second, respectively.”

Supp Fig 8 does include the (photoelectron plus read noise limited) signal-to noise ratio both at the integration light curve level, and the final light-curve fit transit-depth uncertainty both as a function of wavelength. Throughout the paper we homogenized the text to use “counts per second” and added a sentence in the methods to indicate JWST outputs flux units in counts per second.

There are some technical issues that need better explanation: the 1/f noise - what it is and (if known) what causes it needs an explanation. Also what are bias frames and what are they used for, and similarly what are reference pixels? (The paper does a good job of explaining the sample-up-the-ramp non-destructive read process, but the 1/f noise, bias frames, and reference pixels need similar clarity.)

Response:

We refer the referee to Schlawin et al. (2020), and references therein for a more detailed discussion of the source of 1/f noise:

“1/f noise. The JWST HgCdTe detector readout system adds correlated read noise to the digital images. This correlated noise is caused by (1) the readout integrated circuits (ROICs), which have a p-type metal-oxide semiconductor field-effect transistor (PFET) source follower, as well as (2) direct current (DC) biases in the SIDECAR ASIC electronics (Rauscher et al. 2011). This electronic read noise has the property that most of the noise power is concentrated at low frequencies.” Schlawin et al. (2020).

To help clarify the 1/f noise, we added the following sentence to the methods citing Schlawin et al. 2020, *“All reductions correct for 1/f noise: correlated frequency-dependent read noise in the images caused by detector readout and current biases in the electronics [33].”*

In the Fig 1 caption - how large is the linear trend that was removed? (Other investigators may want to compare to data in other programs.)

Response:

The linear trend varies from 0 to -400 ppm per hour as a function of wavelength. The strength of the trend peaks sharply at 3.2 microns. We note that the structure of the linear trend as a function of wavelength does not correspond to any spectral features in the transmission spectrum, or any other systematic vectors. This wavelength-dependent trend warrants a deeper exploration in a future work. We have added the average magnitude of the trend to the caption of Fig. 1.

When fitting the white-light curve, non-Gaussian degeneracies in the posterior distributions are mentioned. Posteriors are mentioned, but is this really a Bayesian fit? MCMC is just a sampler, it doesn't imply that the fitting process is Bayesian. If the fit is Bayesian, what are the priors? (There

can be no posteriors without priors). I don't think a Bayesian fit is needed, but a clear description of the principles of the fit is definitely needed.

Response:

We agree that the language was vague. We have replaced the wording in the 4th paragraph of the FIREFLY subsection in the Methods with this: “We fit this white-light curve using the Markov Chain Monte Carlo sampler emcee within the least-squares minimization framework of lmfit. We use 1,000 steps and uniform priors with extremely wide bounds that encapsulate the limits of physicality to ensure that there is no bias introduced by the prior. Our fitting approach takes into account non-Gaussian degeneracies in the posterior distribution, thereby addressing the known linear correlation between impact parameter (b) and the scaled semimajor axis (a/R_*).”

Figure 5 is a good idea, but the current version isn't useful, because differences are too small to see. Consider subtracting a mean frame and showing the differences on an expanded scale.\

Response:

We have elected to keep Fig 5 in its current form as the main aim is to show the four reductions are very similar, and indeed nearly identical at the light curve level. As the four reductions all reach nearly identical spectra which typically agree at the 1-sigma level, showing residual light curves vs a median subtracted value would over amplify differences which are negligible.

The phrase "used them to shift stabilize the images with flux-conserving interpolation." needs a better explanation. (Stabilizing an image sounds like something that is done by hardware, not by data analysis.)

Response:

We have added this language to clarify this in the third paragraph of the FIREFLY subsection of Methods: “We use cross-correlation along both axes of the cleaned count rate frames to measure the motion of the PSF across the detector through time. To mitigate the effects of this instrumental jitter, we numerically shift each frame with flux-conserving interpolation.”

"We chose a 4-pixel-wide aperture to encompass the peak of the stellar flux without unnecessary background." If background is significant, then an optimally-weighted extraction should be done as per the standard recipe (PASP, 98, 609). If background is negligible compared to this bright star, say so explicitly (and use a wider window to gather maximum signal). Background subtraction is an important technical issue and differences among pipelines should be highlighted in a summary table.

Response:

The background is dominated by zodiacal light, and through our small aperture is very small, estimated to be a few counts per second using the JWST ETC. The text had a poorly worded explanation, which we have updated. In fact there is little background at the integration level because the background is removed by the $1/f$ (column-by-column median) subtraction which is performed at

the group level. However, the Tiberius pipeline additionally fits a column-by-column linear polynomial to fit any residual slopes in the background at the integration level. We experimented with the impacts of this additional step and found that for a 4-pixel-wide aperture, the median uncertainty in the transmission spectrum dropped by 3% after performing this additional step. We also followed the referee's suggestion and ran new reductions with wider apertures with full widths of 8 and 16 pixels (2x and 4x our original aperture). The 8-pixel-wide aperture gives a median uncertainty 1% larger than a 4-pixel aperture and a 16-pixel aperture gives an uncertainty 15% larger than 4-pixels. This same change was reflected in the median RMS of the residuals to the light curve fits. Since the stellar PSF is so narrow in PRISM data, we believe that the increase in noise with increasing aperture width is related to the increasing influence of photon noise, readnoise and bad pixels where the stellar flux is lower. Given these tests, we retain the same aperture width and treatment of the background as in the first version of the paper.

We have updated the Tiberius section of the methods to convey these tests, and have added clarification of background and $1/f$ subtraction in the new Table 2 for each of the reductions.

"We calculate uncertainties in the stellar spectra from the photon noise (before the background subtraction step)" That statement raises questions about precisely how that was done, and was it a correct procedure. Please explain more clearly, using an equation if needed.

Response:

This indeed was not accurate as we were initially calculating the uncertainties on the post $1/f$ -subtracted data which is the equivalent of subtracting a background and then estimating the photon noise. We have now removed this sentence as it is an incorrect procedure and the stellar flux uncertainties do not make a difference to our analysis. This is because the light curve uncertainties are rescaled during our fitting process to give a reduced $\chi^2 = 1$ before a subsequent fit. We tested this by refitting our light curves with photometric uncertainties initially set to unity (i.e. the uncertainties had no weight) and we got an identical transmission spectrum with the same uncertainties.

Regarding Figure 9 and the statement "This offset suggests the limb of WASP-39A is brighter than the stellar models predict." Please re-visit this in more depth. Typically, quadratic functions don't fit model atmosphere intensities very well near the limb. So if they don't fit models very well, then of course they won't fit the real star very well either. I'm suggesting that the problem could be with the quadratic approximation, not the model atmospheres. Showing a quadratic fit to the model's center-to-limb intensities could help to clarify this issue.

Response:

Since the original submission, the Firefly light curve fits have been updated and a correction was made regarding the eccentricity which was unintentionally set at a small value of 0.05. A small eccentricity is degenerate with the shape of the stellar limb darkening, resulting in a larger coefficient offset than if the eccentricity is set to zero, which is the expected value for this planet. When setting the eccentricity to zero, the shape and differential transit depths of the transmission spectrum are

identical as the non-zero value; however, there is an offset which lowers all the transit depth values by 69 ppm. We have updated the manuscript to correct this, though the main figures appear identical, as a normalization offset was already applied to each reduction to remove similar offsets resulting from slightly different system parameters and limb darkening prescriptions. However, the limb-darkening figure in the Methods now shows a much smaller difference between the fit LD coefficients and the stellar models, with only a ~6% difference in the $u+$ term.

Regarding the use of the quadratic limb darkening law and not fitting the limbs well, in principle we fully agree with the referee. However, when one inspects the empirically derived coefficients the $u-$ term fits to smaller values, and is consistent with zero longward of 1 to 2 microns indicating the data prefer a largely linear stellar intensity profile.

In the caption of Figure 8, it would be helpful to subsequent investigators to be more specific concerning the properties of the noise. Instead of saying "The residual scatter is approximately Gaussian", please be quantitative and quote the results of Anderson-Darling tests, or similar normality statistics.

Response:

We performed the suggested statistical test, finding the residuals are consistent with a Gaussian distribution, the following text was added to the caption of Fig 8:

"The residual scatter is approximately Gaussian for each, as indicated by the histogram on the right y-axis. We validate this by performing Anderson-Darling tests on the residuals of the spectral and white-light curves, and find that all of the Anderson-Darling test statistics lie below the respective critical values 1\% significance level. Therefore, we find that there is not sufficient evidence that the residuals are not normally distributed."

When describing the tshirt spectral extractions, the statement "No corrections were made to the centroid or wavelength solution due to the exceptional pointing stability of the observatory" seems to contradict a previous statement that "We measure $x-$ and $y-$ jitter systematics at the ~100 ppm level." I realize that different pipelines will make different approximations, but either the X- and Y- shifts are significant for extracting the transit spectrum, or they're not. Please be clear about whether position-related corrections are needed. It would be helpful to plot the X and Y shifts as a function of time - you already have 25 time series plots on Fig 2, so I'm confident you can find room to plot X- and Y- time series somewhere. And please describe (in a summary table) how different pipelines treat this issue.

Response:

From the FIREFLY reduction, we find correcting for the centroid can reduce the scatter of the residuals. However, in this case the X,Y centroid values themselves have no apparent drift over the observations and appear to randomly 'jitter' around a central X,Y detector location. Thus, there is little to no change in the transit depth values themselves when correcting for the centroid, and the Tshirt reduction elected not to correct for these shifts. We have clarified the use of this issue in the Methods Table and have added the X and Y shifts as a function of time to Methods Figure 8.

In the tshirt section, "We also assume an exponential ramp in time to the data", be careful using the word "ramp". The Spitzer investigators used it to mean a temporal baseline spanning much of the transit. JWST detectors use a sample-up-the-ramp procedure, and that's a different usage of ramp. I suggest saying "temporal baseline" when that's what you mean. (The term "ramp" wasn't confusing in the Spitzer era, but it could be now.)

Response:

This is a good point, we have reworded the statement in the manuscript to read “*We also assume an exponential temporal baseline in time to the data...*”

In the Tiberius section, you describe "replacing pixels containing a cosmic ray with a linear interpolation between neighboring pixels." Strictly speaking, replacing pixels is an incorrect procedure (you're manufacturing data that you don't have, based on unverified assumptions). The strictly correct procedure is to zero-weight the bad pixels. In actual practice, it probably doesn't matter, but it would be helpful to describe some tests to ensure that your transit spectrum is not sensitive to this procedure.

Response:

We performed a test with zero-weighting the pixels and indeed found that this resulted in a transmission spectrum that was consistent to $\ll 1$ sigma with a median uncertainty that differed by 1 ppm.

As regards your best-fit grey cloud opacity, it is very interesting and would be worth noting that the value you derive is close to the grey opacity inferred by early HST spatial scan spectroscopy of water vapor in HD209458b ($0.012 \text{ cm}^2/\text{g}$, see ApJ, 774, 95).

solar should not be capitalized. The Sun is capitalized, but not solar.

Response:

We have corrected the capitalization of solar in the manuscript.

Reviewer Reports on the First Revision:

Referee #2 (Remarks to the Author):

The authors have done a good job responding to my extensive comments on the first version. I think the revisions are mostly fine, and I have only two remaining issues. I am prepared to quickly recommend publication in Nature if the authors can respond to these points:

1. I think my concern in "Issue 5" hasn't yet been fully addressed. I agree with the procedure of using a Gaussian/Voigt fit "to determine if there is a significant signal present without having to identify the molecular absorber." And when the center of that fit agrees with the centroid of opacity for a given molecule such as SO₂, then a tentative identification of that molecule is reasonable. But the data seem good enough to allow more than just a centroid analysis: the data could in principle reject an identification based on shape. For example the modeled SO₂ absorption might be asymmetric with a centroid that agrees with the data, but with a shape that clearly disagrees. Probably the authors have compared the data to modeled spectra for SO₂ and have verified that the band shape isn't a problem. In that case the paper needs an explicit statement saying (at a minimum) that the modeled shape of the SO₂ absorption is not in obvious disagreement with the data. (Or perhaps the authors haven't investigated the band shape, in which case that should be stated clearly.)

2. I should have noticed this in the first round: when quoting transit times in BJD, please add TDB. Almost nobody quotes BJD(UTC) anymore, but it's best to be explicit and say BJD(TDB).

Author Rebuttals to First Revision:

1. I think my concern in "Issue 5" hasn't yet been fully addressed. I agree with the procedure of using a Gaussian/Voigt fit "to determine if there is a significant signal present without having to identify the molecular absorber." And when the center of that fit agrees with the centroid of opacity for a given molecule such as SO₂, then a tentative identification of that molecule is reasonable. But the data seem good enough to allow more than just a centroid analysis: the data could in principle reject an identification based on shape. For example the modeled SO₂ absorption might be asymmetric with a centroid that agrees with the data, but with a shape that clearly disagrees. Probably the authors have compared the data to modeled spectra for SO₂ and have verified that the band shape isn't a problem. In that case the paper needs an explicit statement saying (at a minimum) that the modeled shape of the SO₂ absorption is not in obvious disagreement with the data. (Or perhaps the authors haven't investigated the band shape, in which case that should be stated clearly.)

With our second method for gas detection, we use molecular opacities rather than Gaussian profiles. In the case of SO₂, this method yields a stronger detection significance than the Gaussian profile. This lends further confidence to the identification and detection of the molecule, as the absorption feature is better fit by the SO₂ opacity profile.

We have added this short paragraph to the end of the "Detection Significance of Gases" section:

"As shown in Table 4, the detection significance of all gases increases with the Bayes factor analysis technique relative to the Gaussian/Voigt function technique. This is notably also the case for SO₂, lending confidence to the detection and identification of the molecule, as the feature is better fit by its respective opacity profile."

2. I should have noticed this in the first round: when quoting transit times in BJD, please add TDB. Almost nobody quotes BJD(UTC) anymore, but it's best to be explicit and say BJD(TDB).

We have made the change to BJD_{TDB} where relevant.